# The transmission blocking activity of artemisinin-combination, non-artemisinin, and 8-aminoquinoline antimalarial therapies: A pooled analysis of individual participant data

Leen N. Vanheer[1,2‡], Jordache Ramjith[3‡], Almahamoudou Mahamar[4‡], Merel J. Smit[3], Kjerstin Lanke[3], Michelle E. Roh[5], Koualy Sanogo[4], Youssouf Sinaba[4], Sidi M. Niambele[4], Makonon Diallo[4], Seydina O. Maguiraga[4], Sekouba Keita[4], Siaka Samake[4], Ahamadou Youssouf[4], Halimatou Diawara[4], Sekou F. Traore[4], Roly Gosling[6,7], Joelle M. Brown[7], Chris Drakeley[1,2], Alassane Dicko[4‡], Will Stone[1,2‡], Teun Bousema[3‡*]

1 Department of Infection Biology, London School of Hygiene & Tropical Medicine, London, United Kingdom, 2 Malaria Centre, London School of Hygiene & Tropical Medicine, London, United Kingdom, 3 Department of Medical Microbiology and Radboud Center for Infectious Diseases, Radboud University Medical Center, Nijmegen, the Netherlands, 4 Malaria Research and Training Centre, Faculty of Pharmacy and Faculty of Medicine and Dentistry, University of Sciences Techniques and Technologies of Bamako, Bamako, Mali, 5 Institute for Global Health Sciences, University of California, San Francisco, California, United States of America, 6 Department of Disease Control, London School of Hygiene and Tropical Medicine, London, United Kingdom, 7 Department of Epidemiology and Biostatistics, University of California, San Francisco, California, United States of America

‡ LNV, JR, and AM are the joint first authors, contributed equally. AD, WS, and TB are the joint last authors, contributed equally.
* teun.bousema@radboudumc.nl

## Abstract

### Background

Interrupting human-to-mosquito transmission is important for malaria elimination strategies as it can reduce infection burden in communities and slow the spread of drug resistance. Antimalarial medications differ in their efficacy in clearing the transmission stages of *Plasmodium falciparum* (gametocytes) and in preventing mosquito infection. Here, we present a retrospective combined analysis of six trials conducted at the same study site with highly consistent methodologies that allows for a direct comparison of the gametocytocidal and transmission-blocking activities of 15 different antimalarial regimens or dosing schedules.

### Methods and findings

Between January 2013 and January 2023, we conducted six clinical trials evaluating antimalarial treatments with transmission endpoints at the Clinical Research Centre of the Malaria Research and Training Centre of the University of Bamako in Mali. These trials tested Artemisinin-Combination Therapies (ACTs), non-ACT

**Data availability statement:** Trial data for NECTAR1-4 studies are available through a novel open-access clinical epidemiology database resource, ClinEpiDB (https://clinepidb.org); and code for analysis and the selection of data used for this analysis can be found on https://github.com/leenvh/NECTAR-pooled-analysis.

**Funding:** This work and the individual clinical trials were supported by the Bill & Melinda Gates Foundation (https://www.gatesfoundation.org, grant numbers OPP1013179 and OPP1089413 to AD, TB and RG, INV-002098 and INV-005735 to AD, CD, WS and TB); LNV was further supported by a Biotechnology and Biological Sciences Research Council LIDo Ph.D. studentship (https://www.ukri.org/councils/bbsrc, reference BB/T008709/1). WS was supported by a Wellcome Trust fellowship (https://wellcome.org, reference 218676/Z/19/Z/WT). JR and TB were funded by a VICI fellowship from the Dutch Research Council (NWO; https://www.nwo.nl; grant number 09150182210039). The funders had no role in study design, data collection and analysis, decision to publish, or preparation of the manuscript.

**Competing interests:** I have read the journal's policy and the authors of this manuscript have the following competing interests: CD is a member of the Malaria Policy Advisory Group (MPAG). TB contributed to a Guideline Development Group of the World Health Organization for recommendations for the final phase of elimination and prevention of re-establishment of malaria.

**Abbreviations:** ACT, artemisinin-based combination therapy; AL, artemether-lumefantrine; AL-AQ; artemether-lumefantrine-amodiaquine; AQ, amodiaquine; AS-AQ, artesunate-amodiaquine; DHA-PPQ, dihydroartemisinin-piperaquine; G6PD; glucose-6-phosphate dehydrogenase; GLMM, generalised linear mixed model; NMA, network meta-analysis; PY-AS, pyronaridine-artesunate; PQ, primaquine; RT-qPCR, reverse-transcriptase quantitative PCR; SLD, single low-dose; SP-AQ, sulfadoxine-pyrimethamine plus amodiaquine; TACT, triple artemisinin-based combination therapy; TQ, tafenoquine; WHO, World Health Organisation.

regimens and combinations with 8-aminoquinolines. Participants were males and non-pregnant females, between 5 and 50 years of age, who presented with *P. falciparum* mono-infection and gametocyte carriage by microscopy. We collected blood samples before and after treatment for thick film microscopy, infectivity assessments by mosquito feeding assays and molecular quantification of gametocytes. To combine direct and indirect effects of treatment groups across studies, we performed a network meta-analysis. This analysis quantified changes in mosquito infection rates and gametocyte densities within treatment groups over time and between treatments. In a pooled analysis of 422 participants, we observed substantial differences between antimalarials in gametocytocidal and transmission-blocking activities. Artemether-lumefantrine (AL) was significantly more potent at reducing mosquito infection rates within 48 h than dihydroartemisinin-piperaquine ($p = 0.0164$) and sulfadoxine-pyrimethamine plus amodiaquine ($p = 0.0451$), while this difference was near-significant for artesunate-amodiaquine ($p = 0.0789$) and pyronaridine-artesunate ($p = 0.0519$). The addition of single low-dose primaquine (SLD PQ) accelerated gametocyte clearance for any ACT and led to a substantially greater reduction in mosquito infection rate within 48 h of treatment for all ACTs except AL, while an SLD of the 8-aminoaquinoline tafenoquine showed a delayed activity, compared to SLD PQ, but was similarly effective. The main limitations of the study include the inclusion of highly infectious individuals, which may not reflect the broader malaria patient population with lower or undetectable gametocyte densities and the small sample sizes in some treatment groups, which resulted in wide confidence intervals and reduced the certainty of effect estimates.

## Conclusions

We found marked differences among ACTs and single low-dose 8-aminoquinoline drugs in their ability and speed to block transmission. The findings from this analysis can support treatment policy decisions for malaria elimination and be integrated into mathematical models to improve the accuracy of predictions regarding community transmission and the spread of drug resistance under varying treatment guidelines.

### Author summary
### Why was this study done?

- Interrupting malaria transmission from humans to mosquitoes is critical for malaria elimination and controlling the spread of drug-resistant parasites.

- Antimalarial drugs differ in their ability to clear gametocytes (the parasite stage responsible for transmission) and block onward transmission to mosquitoes, but comparative evidence has been limited.

- We aimed to provide a comprehensive comparison of the gametocytocidal and transmission-blocking efficacy of anti-malarial treatments that are currently used or considered for first-line treatment or chemoprevention, using data from six clinical trials conducted at a single site over a 10-year period.

## What did the researchers do and find?

- We conducted a network meta-analysis using individual participant data from six clinical trials with standardised mosquito feeding assays and gametocyte quantification before and after treatment to compare 15 antimalarial regimens.

- We found substantial differences between treatments in how quickly and effectively they reduced gametocyte levels and mosquito infectivity.

- Artemether-lumefantrine was more effective than several other standard-dose anti-asexual drugs at reducing mosquito infection within 48 h. Adding single low-dose primaquine rapidly enhanced transmission-blocking, while a single low-dose tafenoquine had a slower but effective impact.

## What do these findings mean?

- The ability and speed with which antimalarials block transmission varies substantially among treatment regimens.

- These findings can guide national treatment policies aimed at reducing malaria transmission and inform mathematical models to predict the community-level impact of different treatment strategies on malaria control and resistance spread.

- Study limitations include that the study participants were highly infectious individuals, who may not represent most malaria patients; in addition, some treatment groups were small, limiting the ability to draw robust conclusion for those groups

## Introduction

The primary aim in the therapeutic management of malaria is the clearance of pathogenic asexual blood-stage parasites, using antimalarials with anti-asexual properties. Artemisinin-based combination therapies (ACTs) are the first-line treatment for uncomplicated *Plasmodium falciparum* malaria across the world and are highly potent against asexual parasites, capable of reducing the circulating asexual parasite biomass ~10,000 fold per 48-h cycle [1]. Gametocytes are distinct parasite stages that are not associated with symptoms but are the only parasite life stage that can be transmitted to and establish infection in mosquitoes. In *P. falciparum*, gametocytes develop during five developmental stages over a prolonged 10–12-day maturation period. Importantly, *P. falciparum* gametocytes have markedly different sensitivity profiles to antimalarial drugs than asexual parasites. Assessing and comparing the gametocytocidal and transmission-blocking properties of different antimalarial regimens is important for informing treatment guidelines that aim to contribute to transmission reduction.

Artemisinins are active against developing gametocytes but less so against the mature forms [2]. Although gametocyte density generally decreases more after treatment with ACTs compared to non-ACTs [3], there is considerable variation in gametocyte-clearing potential between ACTs [4]. For example, current artemisinin partner drugs such as piperaquine, lumefantrine and amodiaquine have limited activity against *P. falciparum* gametocytes at clinically relevant concentrations [2,5–7]. In vitro data for the partner drug pyronaridine are contradictory; one study found a strong effect against mature gametocytes [8], whereas others only observed activity against mature gametocytes at concentrations above the therapeutic threshold [2,6,9]. Furthermore, a disconnect exists between the detection of gametocytes and infectivity to

mosquitoes; on the one hand, gametocytes can be infectious at sub-microscopic densities [10], whereas on the other, antimalarials can have a parasite-inhibiting effect, active after ingestion by mosquitoes [2,11] or sterilise gametocytes so that these are detectable but not transmissible [12]. Gametocyte quantification is therefore an imperfect measure of post-treatment transmission potential, and mosquito feeding assays are considered more informative [13]. These assays involve feeding mosquitoes on blood from infected individuals and allow for measurement of mosquito infection rate, oocyst density, and the proportion of individuals who are infectious.

With the threat of artemisinin partial resistance in sub-Saharan Africa [14], various strategies have been proposed to counter the spread of resistant parasites. One of the most promising approaches, according to modelling simulations, is the use of Triple Artemisinin-based Combination Therapies (TACT), that combine existing ACTs with a second partner drug. This approach may offer protection against partner drug resistance and maintain high treatment efficacy in areas where resistance against artemisinins and one of the partner drugs is present [15]. Other suggested strategies include multiple first-line ACT therapies or cycling between different ACTs [16,17]. An alternative or complementary strategy is to supplement treatment with gametocytocidal compounds. Primaquine (PQ) and its long-lasting analogue tafenoquine (TQ) are 8-aminoquinolines that can clear *P. vivax* liver stages [18] and have *P. falciparum* gametocytocidal activity. The WHO recommends supplementing first-line ACTs with a single low-dose (0.25 mg/kg) PQ (SLD PQ) in low transmission areas [19], and the WHO malaria policy and advisory group has now suggested expanding this recommendation to other regions [20]. TQ holds the promise of long-lasting transmission blocking activity, and recently, a single low dose of TQ (SLD TQ) in combination with ACTs and non-ACT treatments was found to effectively block transmission within 7 days at a dose of 1.66 mg/kg [21,22]. To date, no trials directly comparing SLD TQ to SLD PQ have been conducted, largely due to the impracticality of trials involving mosquito feeding assays.

In order to make informed decisions regarding optimal antimalarial regimens, it is necessary to understand and compare the transmission-blocking abilities of different ACTs and TACTs. We conducted a pooled analysis of individual-level data from six clinical trials to compare the transmission-blocking effects of 15 different antimalarials and combinations. All six trials measured mosquito transmission endpoints, used sensitive gametocyte quantification, and were conducted using the same mosquito feeding protocols and assays in Ouélessébougou, Mali, between 2013 and 2023 [21–26]. Results from this analysis can be used in mathematical models to more accurately predict community transmission and the spread of drug resistance under different treatment guidelines, and will inform malaria control programmes.

## Methods

### Study design and participants

Between January 2013 and January 2023, six clinical trials [21–26] (Table 1 and S2 Appendix) involving a total of 521 participants were performed at the Clinical Research Centre of the Malaria Research and Training Centre (MRTC) of the University of Bamako (Bamako, Mali). Study participants were recruited in the commune of Ouélessébougou, which includes the central town of Ouélessébougou and 44 surrounding villages, with an estimated 50,000 inhabitants and located approximately 80 km south of Bamako, the capital of Mali. Malaria transmission in this region is hyperendemic and highly seasonal with incidence peaking during and following the rainy season from July to November. In all six trials, participants were males and non-pregnant females, between 5 and 50 years of age, with a body weight less than 80 kg, who presented with *P. falciparum* mono-infection and at least 1 (study acronyms PQ03, NECTAR 1-4) or 2 (study PQ01) gametocytes per 500 white blood cells (WBC) detected by blood smear. This minimum gametocyte density corresponds to an approximate minimum of 16–32 gametocytes per μL of blood when assuming 8,000 WBC per μL. Participants in all studies, except for the 2013–2014 PQ01 study, were exclusively asymptomatic, and four studies (study acronyms PQ01, PQ03, NECTAR2, NECTAR3) required participants to have normal glucose-6-phosphate dehydrogenase (G6PD) production, as determined by OSMMR-D-D calorimetric test (R&D Diagnostics, Aghia Paraskevi, Greece; studies PQ01, NECTAR2 and NECTAR3), CareStart G6PD rapid diagnostic test (Access Bio, Somerset, NJ, USA; study PQ03) and/

**Table 1. Overview of the included studies. Rows in grey indicate treatment groups that were not included in the analyses.**

| Study | Year | Treatment group | Number of participants | Treatment category | Study population | Reference |
|---|---|---|---|---|---|---|
| PQ01 | 2013-2014 | DHA-PPQ | 16 | DHA-PPQ | G6PD+ males, 5–50 years, ≥32 gametocytes/μL | [24] |
| | | DHA-PPQ + 0.0625 mg/kg PQ | 16 | / | | |
| | | DHA-PPQ + 0.125 mg/kg PQ | 17 | / | | |
| | | DHA-PPQ + 0.25 mg/kg PQ | 15 | ACT-PQ | | |
| | | DHA-PPQ + 0.5 mg/kg PQ | 17 | ACT-PQ | | |
| PQ03 | 2016 | SP-AQ | 20 | SP-AQ | G6PD+ males, 5–50 years, asymptomatic, ≥16 gametocytes/μL | [23] |
| | | SP-AQ + 0.25 mg/kg PQ | 20 | Non-ACT-PQ | | |
| | | DHA-PPQ | 20 | DHA-PPQ | | |
| | | DHA-PPQ + MB | 20 | / | | |
| NECTAR1 | 2019 | DHA-PPQ | 25 | DHA-PPQ | Males and non-pregnant females, 5–50 years, asymptomatic, ≥16 gametocytes/μL | [25] |
| | | DHA-PPQ + 0.25 mg/kg PQ | 25 | ACT-PQ | | |
| | | PY-AS | 25 | PY-AS | | |
| | | PY-AS + 0.25 mg/kg PQ | 25 | ACT-PQ | | |
| NECTAR2 | 2020 | DHA-PPQ | 20 | DHA-PPQ | G6PD+ males and non-pregnant females, 12–50 years, asymptomatic, ≥16 gametocytes/μL | [22] |
| | | DHA-PPQ + 0.42 mg/kg TQ | 20 | / | | |
| | | DHA-PPQ + 0.83 mg/kg TQ | 20 | ACT-TQ | | |
| | | DHA-PPQ + 1.66 mg/kg TQ | 20 | ACT-TQ | | |
| NECTAR3 | 2021 | AL | 20 | AL | G6PD+ males and non-pregnant females, 10–50 years, asymptomatic, ≥16 gametocytes/μL | [21] |
| | | AL + 0.25 mg/kg PQ | 20 | ACT-PQ | | |
| | | SP-AQ | 20 | SP-AQ | | |
| | | SP-AQ + 1.66 mg/kg TQ | 20 | / | | |
| NECTAR4 | 2022 | AL | 20 | AL | Males and non-pregnant females, 10–50 years, asymptomatic, ≥16 gametocytes/μL | [26] |
| | | AL-AQ | 20 | AL | | |
| | | AL-AQ + 0.25 mg/kg PQ | 20 | ACT-PQ | | |
| | | AS-AQ | 20 | AS-AQ | | |
| | | AS-AQ + 0.25 mg/kg PQ | 20 | ACT-PQ | | |

or STANDARD G6PD quantitative enzyme activity test (SD Biosensor, Suwon, South Korea; studies NECTAR2 and NECTAR3; Table 1 and S2 Appendix). Other study-specific inclusion criteria are presented in S2 Appendix. The sample size of individual studies was based on achieving ≥80% power to detect a 90% reduction in the proportion infected mosquitoes on, depending on the drug regimen, day 2 or 7. Before screening and before study enrolment, participants provided written informed consent (if they were aged ≥18 years) or written parental consent for participants younger than 18 years. In addition to parental consent, oral assent was sought for individuals aged 12–17 years in studies PQ01, PQ03, NECTAR1-2 and 10–17 years in studies NECTAR3-4. Ethical approvals for the individual studies were obtained from the Ethics Committee of the Faculty of Medicine, Pharmacy, and Dentistry of the University of Science, Techniques, and Technologies of Bamako (Bamako, Mali). In addition, the studies were approved by the Committee on Human Research at the University of California (San Francisco, CA, USA), and/or the Research Ethics Committee of the London School of Hygiene and Tropical Medicine (London, UK).

**Procedures**

Of the antimalarial treatments evaluated in the included studies, the following were included in this analysis: (i) Dihydroartemisinin-piperaquine (DHA-PPQ; Eurartesim; Sigma Tau, Gaithersburg, MD, USA); (ii) Pyronaridine-artesunate (PY-AS; Pyramax; Shin Poong Pharmaceutical, Seoul, South Korea; (iii) Artemether-lumefantrine (AL; Coartem; Novartis,

Basel, Switzerland or Guilin Pharmaceutical, Shanghai, China); (iv) Artesunate-amodiaquine (AS-AQ; Guilin Pharmaceutical, Shanghai, China); (v) Sulfadoxine-pyrimethamine plus amodiaquine (SP-AQ; Guilin Pharmaceutical, Shanghai, China); (vi) Any of the ACTs listed above plus a single low-dose of primaquine (ACT-PQ; 0.25 or 0.5 mg/kg; Sanofi, Laval, QC, Canada or ACE Pharmaceuticals, Zeewolde, the Netherlands); (vii) Non-ACT (SP-AQ) plus a single low-dose of primaquine (Non-ACT-PQ; Sanofi, Laval, QC, Canada); (viii) ACT (DHA-PPQ) plus a single low dose of tafenoquine (ACT-TQ; 0.83 or 1.66 mg/kg; 60° Pharmaceuticals Ltd, USA) (Table 1). Antimalarial treatments were administered as per manufacturer's instructions (Text A for dosing tables and Table A for manufacturers in S1 Appendix, pp 3–6) under direct supervision. ACT treatments and sulfadoxine-pyrimethamine plus amodiaquine were administered over 3 days (days 0, 1, and 2). Primaquine and tafenoquine were administered as a single dose immediately after the first dose of ACT. In the NECTAR1 study, participants were treated with a full course of DHA-PPQ at day 21 of follow-up, to prevent re-infection. For the current analysis, we did not consider study arms within the included studies that are currently not considered as treatment regimens, such as arms with PQ doses below 0.25 mg/kg ($n = 33$ participants) and with TQ doses below 0.83 mg/kg ($n = 20$) or treatment arms involving methylene blue ($n = 20$). The combination of SP-AQ with an SLD TQ ($n = 20$) was also not considered in the analysis since SP-AQ is currently only considered for seasonal malaria chemoprevention (SMC) and the addition of SLD TQ is not considered for this purpose. In all studies, blood samples were taken before treatment and on days 2 and 7 following treatment for thick film microscopy, infectivity assessments and molecular analysis of gametocyte density, as outlined in Fig 1 and described in detail in S2 Appendix.

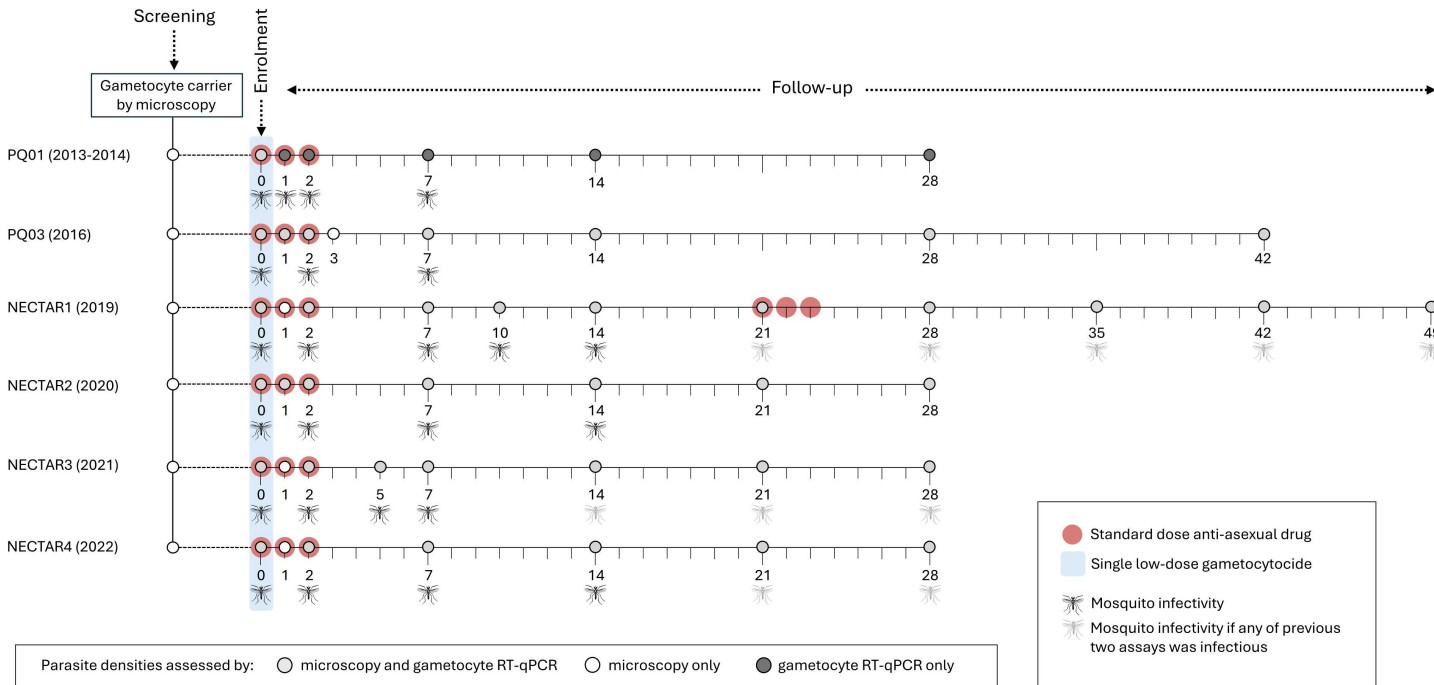

**Fig 1. Study design of included trials.** All studies assessed infectivity to mosquitoes before (d0), during (d2) and one week after initiation of treatment (d7) with additional time-points that differed between studies. Tick marks represent days; circles indicate sampling/screening time points. Circles that are encompassed by larger red circles indicate that a standard dose anti-asexual antimalarial was administered at these study visits, while a single low-dose gametocytocide was administered immediately after the first dose of anti-asexual treatment (marked in blue). Grey coloured circles indicate that parasite densities were assessed by both microscopy (asexual parasite and gametocyte densities) and RT-qPCR (gametocyte densities). At other study visits, parasite densities were determined by microscopy only (white circles) or RT-qPCR only (dark grey circles). Mosquito infectivity assays were conducted at the study visits marked with a black mosquito symbol, while a grey mosquito symbol indicates that mosquito infectivity assays were only conducted at that time point if any of the previous two assays resulted in at least one infected mosquito. Mosquito image in this figure is adapted from https://openclipart.org/detail/233084/mosquito, available under a Creative Commons Zero 1.0 Public Domain License.

For infectivity assessments in all studies, 75–90 insectary-reared *Anopheles gambiae* females were allowed to feed for 15–20 min on venous blood samples collected in Lithium Heparin tubes and offered in a water-jacketed membrane feeder system, as previously described [27]. Fully fed mosquitoes were kept in a temperature and humidity-controlled insectary until day 7 post-feeding, and then dissected in 1% mercurochrome to detect and quantify *P. falciparum* oocysts. Giemsa-stained blood slides were double read by expert research microscopists with asexual stages counted against 200 white blood cells and gametocytes counted against 500 white blood cells. For molecular gametocyte quantification in all studies, venous or finger-prick blood collected in EDTA tubes was aliquoted into L6 buffer (Severn Biotech, Kidderminster, UK) or RNA protect cell reagent (Qiagen, Hilden, Germany) and stored at ≤−70°C until total nucleic acid extraction using a MagNAPure LC automated extractor (Total Nucleic Acid Isolation Kit High Performance; Roche Applied Science, Indianapolis, IN, USA). Male and female gametocytes were quantified in a multiplex reverse-transcriptase quantitative PCR (RT-qPCR) assay targeting Pfs25 or CCP4 transcripts for female gametocytes and PfMGET for male gametocyte quantification, as specified in S2 Appendix. Molecular gametocyte quantification was repeated for the PQ03 study, after the original density estimates of this study (but not of other studies) showed marked disagreement with their respective microscopy density measurements (Fig A in S1 Appendix, p 7). Samples were classified as negative for a specific gametocyte sex if the RT-qPCR quantified gametocyte density was less than 0.01 gametocytes per μL (equivalent to one gametocyte per 100 μL of the blood sample).

## Statistical analysis

Asexual parasite and microscopy gametocyte density distributions at baseline were presented per study by violin plots. Six individuals from the PQ01 study that lacked baseline infectivity measures or microscopy gametocyte densities were excluded from analysis. The association between gametocyte density (gametocytes/ μL) on a log10 scale and the proportion of infected mosquitoes was determined by a mixed-effects logistic regression with a random effect for study. The agreement between (log10) gametocyte density estimates by microscopy and molecular methods was assessed and presented in a Bland-Altman plot, stratified by study, which was created using the BlandAltmanLeh R package [28].

We evaluated the efficacy of multiple antimalarial treatments across six studies using individual participant data (IPD) to examine both time-to-event outcomes and longitudinal outcomes. The survival endpoints were gametocyte clearance, assessed by microscopy and RT-qPCR, and infectivity clearance. Longitudinal outcomes included gametocyte density (microscopy and RT-qPCR), gametocyte prevalence (microscopy and RT-qPCR), proportion of infected mosquitoes, mean oocyst density, and the likelihood of infecting at least one mosquito. For all outcomes, we implemented a two-stage individual participant data network meta-analysis (IPD-NMA), following guidance from Debray and colleagues [29] and Florez and colleagues [30].

For the survival outcomes, to estimate the hazard ratios (HRs) and their standard error for all pairwise comparisons of treatment groups for the first stage, we fitted a separate Cox proportional hazards model for each of the six studies. For longitudinal outcomes, we were primarily interested in quantifying reductions from baseline and comparing these reductions between treatment groups at day 2, day 7, and day 14. To estimate these changes averaged across studies, we used generalised linear mixed models (GLMMs) using the *mgcv* R package [31], treating time point, treatment group, and their interaction as fixed effects. Random intercepts were included for each participant to account for within-individual correlation. For study-specific effects, as required for NMA, we included an interaction with study as a fixed effect in the model. For each outcome, we extracted the estimated relative reduction from baseline for each treatment group and time point, along with their standard errors. As part of a sensitivity analysis, we also estimated the relative reductions from baseline for ungrouped treatment arms. We then calculated all possible pairwise differences in these reductions between groups within each study. The standard errors for these differences were calculated using the delta method. These study-specific pairwise comparisons of treatment groups, from the survival and longitudinal models, formed the input for the second-stage analysis, i.e., the NMA, to allow the additional comparison of treatment groups that were not directly compared within studies.

The NMA was performed using the *netmeta* package in R [32]. For the longitudinal outcomes, the NMA was performed separately for each time point (day 2, day 7 and day 14). We assessed consistency by comparing the average direct (effect from comparisons made in the same study) and indirect (effect from comparisons made between studies) effect estimates. These were visualised by plotting the mean estimate of direct effect and its confidence interval alongside the mean indirect effect and its confidence interval. The network for each outcome was visualised using a network plot, and comparisons with the reference group ACT-PQ were summarised in forest plots. This reference group was chosen because it was the treatment that was present in most individual studies and therefore allowed for more informative direct effect estimates. Additionally, P-scores were calculated for each treatment as a measure of the extent to which a treatment consistently ranks better than competing treatments, based on the point estimates and standard errors from the network meta-analysis. P-scores range from 0 (least favourable) to 1 (most favourable) and reflect the relative standing of each treatment within the network. This was only done for the proportion infected mosquitoes, as the endpoint of greatest relevance to public health.

Kaplan–Meier survival curves were generated with the *survfit* function from the R-based *survival* package [33] to display the cumulative probability of remaining uncleared of gametocytes detected by microscopy, gametocytes detected by RT-qPCR and any infectivity to mosquitoes. Statistical analyses and visualisations were conducted in R (version 4.4.1).

The original studies were registered with ClinicalTrials.gov (NCT01743820, NCT02831023, NCT04049916, NCT04609098, NCT05081089 and NCT05550909). This study was reported as per the Preferred Reporting Items for Systematic Reviews and Meta-Analyses (PRISMA) guideline (S1 PRISMA Checklist).

## Results

### Baseline characteristics were similar across studies

A total of 422 study participants (42–100 participants per study) and 15 different treatment regimens or dosing schedules were evaluated across the six included trials (20–79 participants per regimen/dosing schedule, Table 1 and S2 Appendix). Parasite density was broadly similar between studies that recruited individuals with an asymptomatic infection (2016–2022) but higher in the PQ01 study conducted in 2013–2014 where clinical malaria patients were eligible (Fig 2A). Although participants were recruited based on microscopy-detected gametocyte carriage, at the time of enrolment (up to 24 h after the initial screening), microscopy gametocyte prevalence was 99.5% (420/422) with a median gametocyte density of 55 (IQR 32–112) gametocytes/μL among participants with detectable gametocytes (Fig 2B). Gametocyte density by microscopy was highly skewed with a range of 15.5–2,720 gametocytes/μL (Table B in S1 Appendix, p 8). Prior to treatment, 68.8% (290/422) of individuals were infectious to mosquitoes (Table B in S1 Appendix, p 8). The proportion of mosquitoes each participant infected at this time point ranged from 0.75% to 94.3% (Fig 2C), which is similar to other studies assessing transmission via direct membrane feeding assays in participants with comparable gametocyte densities [34–36]. The proportion of infected mosquitoes was positively associated with microscopy-determined gametocyte density (Fig 2D). In the four studies where oocyst numbers were recorded (NECTAR studies 1–4), the proportion of mosquitoes infected was strongly positively associated with mosquito infection burden (oocyst density) (Fig B in S1 Appendix, p 9).

### Reductions in gametocyte prevalence and densities differ between treatments

Acknowledging that densities below the microscopy threshold for detection (16 gametocytes/μL) can result in mosquito infections, gametocyte density was also quantified by molecular methods. For all studies, gametocyte prevalence by molecular methods was 100% at enrolment and there was agreement between gametocyte density by microscopy and by molecular methods (Fig 3A). At enrolment, two study participants were positive for gametocytes by molecular methods but not by microscopy; both became gametocyte positive by microscopy one day after baseline. Across all post-treatment timepoints, the proportion of gametocyte infections that were submicroscopic was 28.63% (777/2714).

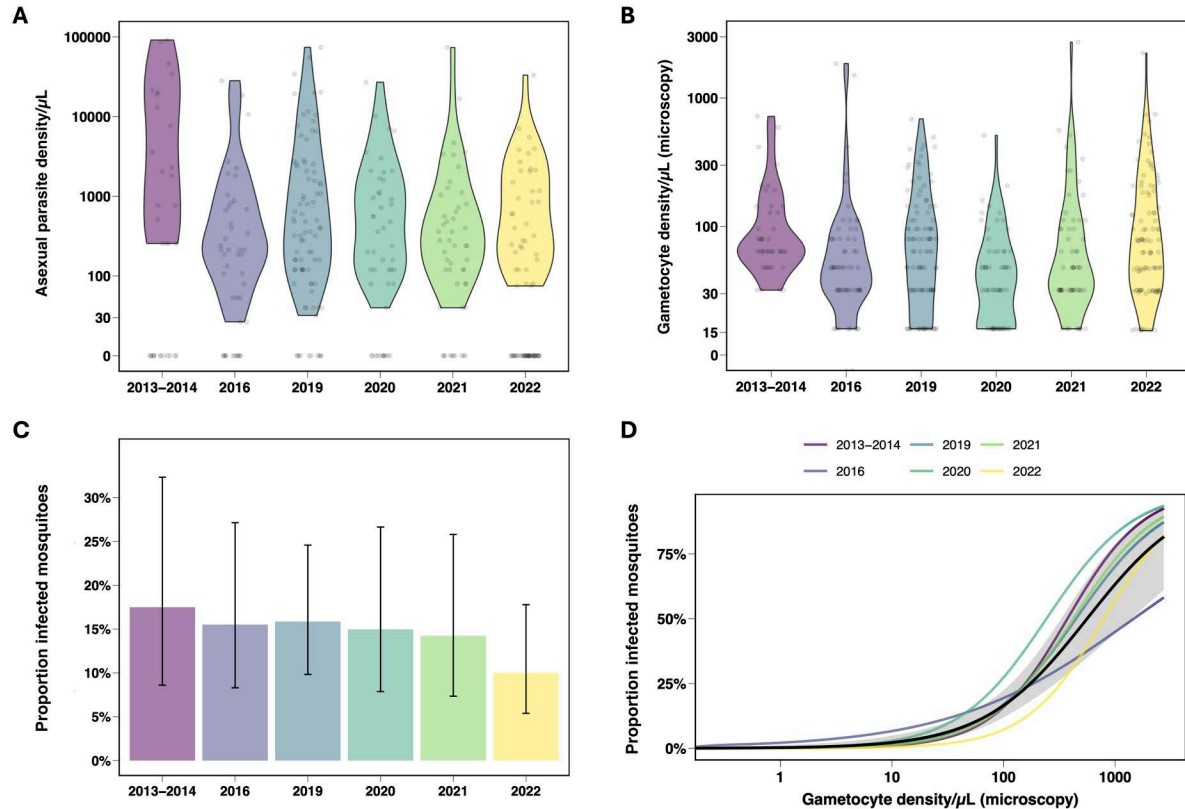

**Fig 2. Baseline study characteristics. A.** Violin plot of microscopy-detected asexual parasite density (parasites/µL) distribution per study. **B.** Violin plot of microscopy-detected gametocyte density (gametocytes/µL) distribution per study. Apparent truncations in panels **A** and **B** reflect characteristics of the data collection process rather than omissions. These arise from (i) inclusion criteria, and (ii) parasite quantification based on reading a fixed fraction of a microliter, which introduces a multiplication factor and results in 'binning' of similar parasite densities. **C.** The mean proportion of mosquitoes that became infected after feeding on venous blood collected at enrolment, prior to treatment, per study. Vertical bars represent 95% *Cis* estimated from a logistic regression model. **D.** Results from a logistic regression between microscopy-determined gametocyte densities (gametocytes/ µL) on a log10 scale and the proportion of infected mosquitoes over the different study years shown by the different colours. Mosquito feeding assays in this analysis were conducted before treatment was initiated. The black line indicates the overall trend averaged across all years with the shaded area showing the 95% confidence interval for this overall fit. Visualisations represent a total of 422 observations, from 42 (2013-2014), 60 (2016), 100 (2019), 60 (2020), 60 (2021) and 100 (2022) study participants. The median number of dissected mosquitoes per study participant (panel C) was 71.8 (IQR 65.6-77) for 2013-2014, 79 (IQR 72-84) for 2016, 64 (IQR 57-70) for 2019, 60 (IQR 51.8-66.5) for 2020, 62 (IQR 53.8-64.2) for 2021 and 61 (IQR 55-66) for 2022.

Following treatment, gametocyte prevalence and densities declined in all treatment groups. The extent and speed at which this occurred depended on the treatment provided (Fig 3B–3D). Two days after treatment, relative reductions compared to baseline in microscopy determined gametocyte prevalence were non-significant in all treatment groups. However, at seven days after treatment initiation, this reduction was statistically significant in the AL group (50.88%, 95% CI 23.07%, 68.63%, *p* = 0.0019), but not in any other group without addition of a gametocytocide (*p* = 0.2169, *p* = 0.2152, *p* = 0.4570 and *p* = 0.7457 after DHA-PPQ, SP-AQ, PY-AS and AS-AQ treatment, respectively). The reduction in gameto-cyte prevalence was most rapid for treatment groups that included a single low-dose gametocytocide, with reductions at day 7 compared to baseline of 68.42% (95% CI 21.76%, 87.25%, *p* = 0.0128) in the non-ACT-PQ group, 90.57% (95% CI 82.08%, 95.03%, *p* < 0.0001) in the ACT-PQ group and 61.54% (95% CI 30.38%, 78.75%, *p* = 0.0016) in the ACT-TQ group. Relative reductions in gametocyte prevalence assessed by microscopy and molecular methods at days 2, 7 and 14 post-treatment are presented in Fig 3B and Table C in S1 Appendix, p 10. We combined different ACTs with PQ for this

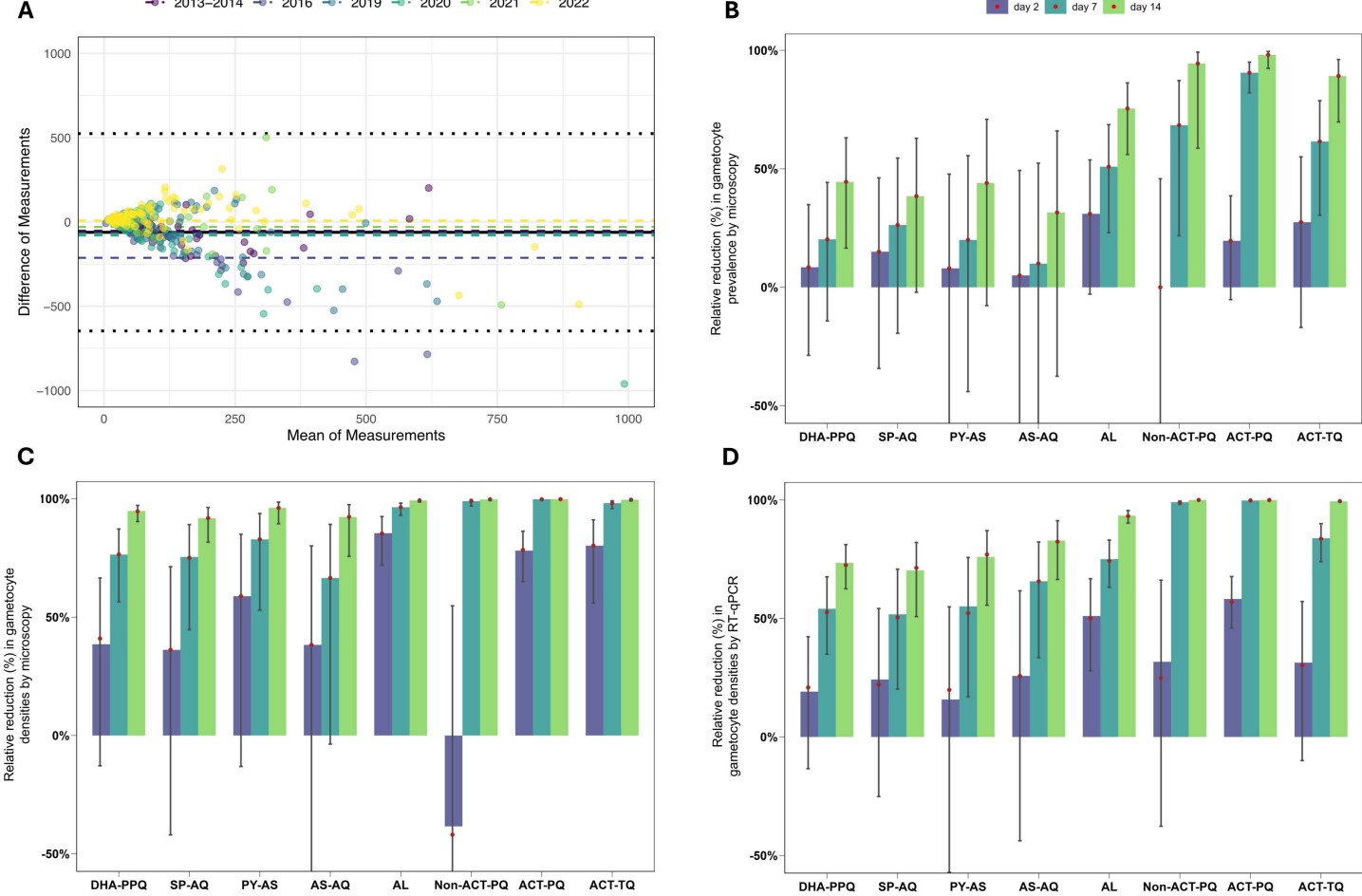

**Fig 3. Gametocyte prevalence and densities. A.** Bland-Altman plot presenting agreement between gametocyte density measured by microscopy and RT-qPCR pre-treatment. The solid black line indicates the overall mean difference including all studies, and dotted lines represent 1.96 × standard deviation of the differences. The dashed lines, coloured by study, represent group-specific mean differences. **B–D.** Bar charts illustrating the relative reduction compared to baseline in gametocyte prevalence by microscopy **(B)**, and gametocyte densities measured by microscopy **(C)** and molecular methods **(D)**, by treatment group over three time points (Day 2 – indigo, Day 7 – turquoise, and Day 14 – green). Vertical bars depict the 95% confidence intervals for these estimates. Red dots represent observed means. The y-axis was cut off below −50 due to inflated standard errors, as a result of reductions from baseline close to zero or high measurement uncertainty. Visualisations represent data from 422 individuals at baseline (79, 40, 25, 20, 60, 20, 138 and 40 individuals from the DHA-PPQ, SP-AQ, PY-AS, AS-AQ, AL, non-ACT-PQ, ACT-PQ and ACT-TQ groups, respectively). At day 2, 369 individuals were included (65, 39, 23, 20, 58, 19, 105 and 40 individuals from the DHA-PPQ, SP-AQ, PY-AS, AS-AQ, AL, Non-ACT-PQ, ACT-PQ and ACT-TQ groups, respectively). Data from 357 individuals at day 7 is shown (57, 38, 24, 20, 57, 19, 104 and 38 individuals from the DHA-PPQ, SP-AQ, PY-AS, AS-AQ, AL, Non-ACT-PQ, ACT-PQ and ACT-TQ groups, respectively). Day 14 includes data from 357 individuals (60, 38, 24, 19, 57, 18, 105 and 36 individuals from the DHA-PPQ, SP-AQ, PY-AS, AS-AQ, AL, Non-ACT-PQ, ACT-PQ and ACT-TQ groups, respectively).

analysis; the findings per study arm are given in Fig C in S1 Appendix, p. 11. An NMA was performed to combine direct and indirect effects of treatment groups across studies and allow for comparisons between treatment groups (Methods A in S1 Appendix, p. 12). Forest plots of treatment comparisons of reduction in gametocyte prevalence by microscopy and molecular methods can be found in Figs D and E in S1 Appendix, pp 13–14.

Reductions in gametocyte density were predictably faster than reductions in gametocyte prevalence; significant relative reductions in density were observed at day 2 compared to baseline by both microscopy and molecular measures in the AL (85.47% (71.88%, 92.49%), $p<0.0001$ and 51.05% (27.92%, 66.76%), $p=0.0003$, respectively) and ACT-PQ (78.12%

(65.02%, 86.32%), $p < 0.0001$ and 58.23% (45.98%, 67.70%), $p < 0.0001$, respectively) groups (Figs 3C–3D and Fig F and Table D in S1 Appendix, pp 15–16).

Reductions in microscopy determined gametocyte density on day 7 after initiation of treatment were more pronounced for AL compared to DHA-PPQ (absolute difference in relative reduction of 47.87% (0.96%, 94.78%), $p = 0.0455$), while this difference was not significant between AL and SP-AQ ($p = 0.2685$), PY-AS ($p = 0.2167$) and AS-AQ ($p = 0.2073$) (Tables E–G in S1 Appendix, pp 17–19). ACT-PQ resulted in the fastest reduction in molecular gametocyte density and this reduction was significantly larger than the best performing ACT, AL, on day 7 and 14 ($p = 0.0010$ and $p = 0.0011$, respectively; Tables H–J and Figs G, H in S1 Appendix, pp 20–24). Non-ACT-PQ and ACT-TQ regimens showed reductions of approximately 30% in molecular gametocyte density by RT-qPCR on day 2, which were substantially smaller than the nearly 60% reduction in the ACT-PQ group, however, these differences were not statistically significant ($p = 0.8144$ and $p = 0.7006$, respectively). ACT-TQ showed a delayed effect on gametocyte density by microscopy compared to ACT-PQ, with a near-significantly smaller relative reduction in the ACT-TQ group compared to the ACT-PQ group at day 7 ($p = 0.0592$), while both treatment groups achieved a >99% reduction in gametocyte density by day 14.

**Reductions in mosquito infection rates differ between artemisinin combination therapies**

A valuable feature of the studies included was the availability of mosquito infection data before and after initiation of treatment. In all groups, 58.3% to 95% of participants infected mosquitoes before treatment (Table B in S1 Appendix, p 8), with 99.7% (289/290) of infectious individuals having gametocytes detected by microscopy at this time point. Following treatment, as gametocyte densities declined, the contribution of submicroscopic infections to transmission increased (Fig 4A). At day 2, 8.33% (12/144) of all infectious individuals had submicroscopic gametocyte infections. At days 7 and 14, these percentages were 6.59% (6/91) and 3.70% (1/27), respectively. The experiments in which mosquitoes became infected after feeding on samples with submicroscopic gametocyte densities typically resulted in low proportions of infected mosquitoes.

Transmission blocking effects per treatment group were quantified as averages of individual-level effects; that is, within-person reductions in the proportion of mosquitoes infected at post-treatment timepoints compared to baseline (Fig 4B). All treatment groups showed a statistically significant relative reduction in mosquito infection rates by day 14 (Fig 4B and Table K in S1 Appendix, p 25). At earlier time-points, we observed a modest but statistically significant reduction in mosquito infection prevalence on day 2 for DHA-PPQ (15.64%, 95% CI 5.15%, 24.97%, $p = 0.0045$) and a greater reduction for non-ACT-PQ (84.44%, 95% CI 79.80%, 88.02%, $p < 0.0001$) whilst high-level transmission reduction (>90%) was observed for AL and for ACTs with SLD PQ. For AS-AQ and ACT with SLD TQ, a statistically significant reduction in mosquito infection rates was only observed at day 7, while for SP-AQ and PY-AS, this was only observed by day 14. The network plot from the NMA showed a sparse structure, with most treatment comparisons supported by only one study (Fig 5A). Ranking the treatment groups based on P-scores identified ACT-PQ, Non-ACT-PQ and AL as the most effective treatments at reducing mosquito infection rates at days 2 and 7 (Fig 5B). Taken together, AL performed significantly better at reducing mosquito infection rate at day 2 of treatment compared to DHA-PPQ ($p = 0.0164$) and SP-AQ ($p = 0.0451$), and near-significantly better compared to PY-AS ($p = 0.0519$) and AS-AQ ($p = 0.0789$) (Fig 5C). By day 7, reductions in the proportion of infected mosquitoes had become more pronounced across most treatments (Figs 5D and Figs I–K in S1 Appendix, pp 26–28).

Over all treatment groups, 27 individuals became infectious to mosquitoes after treatment while initially not infecting mosquitoes; 13.92% (11/79) of participants in DHA-PPQ groups, 12.50% (5/40) in SP-AQ groups, 16.00% (4/25) in the PY-AS group, 10% (2/20) in the AS-AQ group, 1.67% (1/60) in AL groups, 0.72% (1/138) in the ACT plus PQ groups and 7.50% (3/40) in ACT plus TQ groups. Predictably, the probability of infecting at least one mosquito was reduced more slowly than the mosquito infection rate, with only the groups with PQ reaching >90% reduction in this probability at day 2, followed by AL that achieved >80% reduction at day 2 and ACT-TQ that achieved >90% reduction at day 7 (Figs L–N and

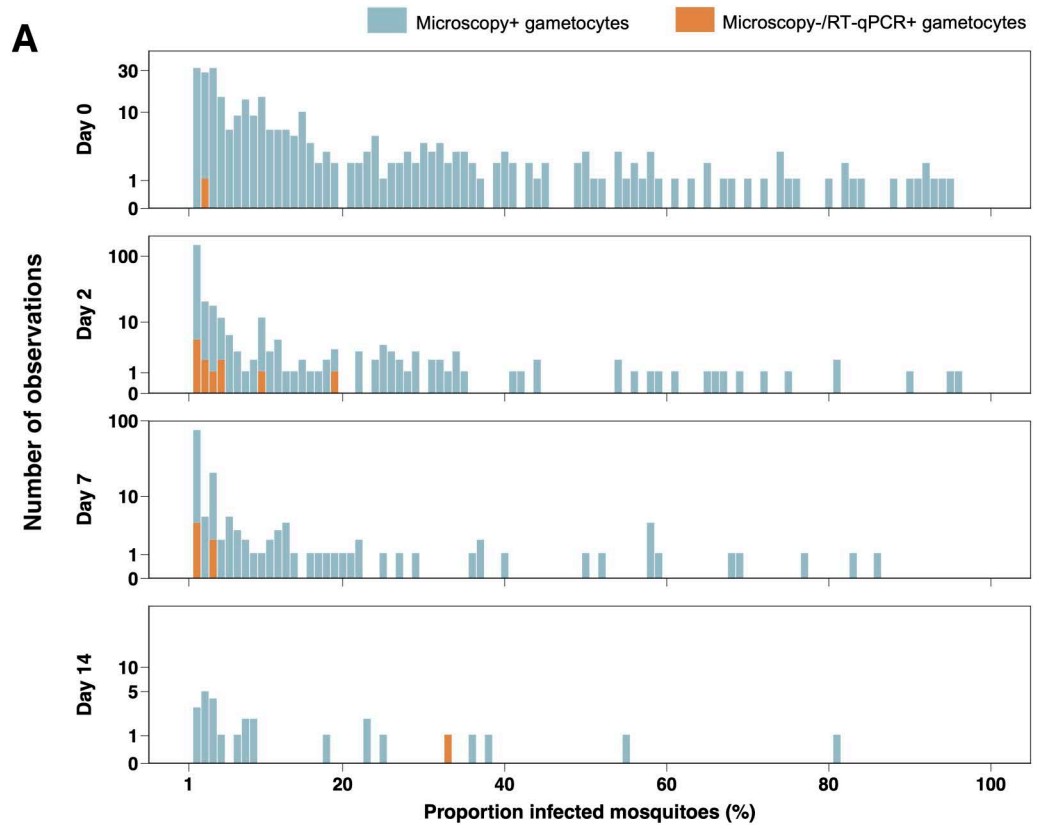

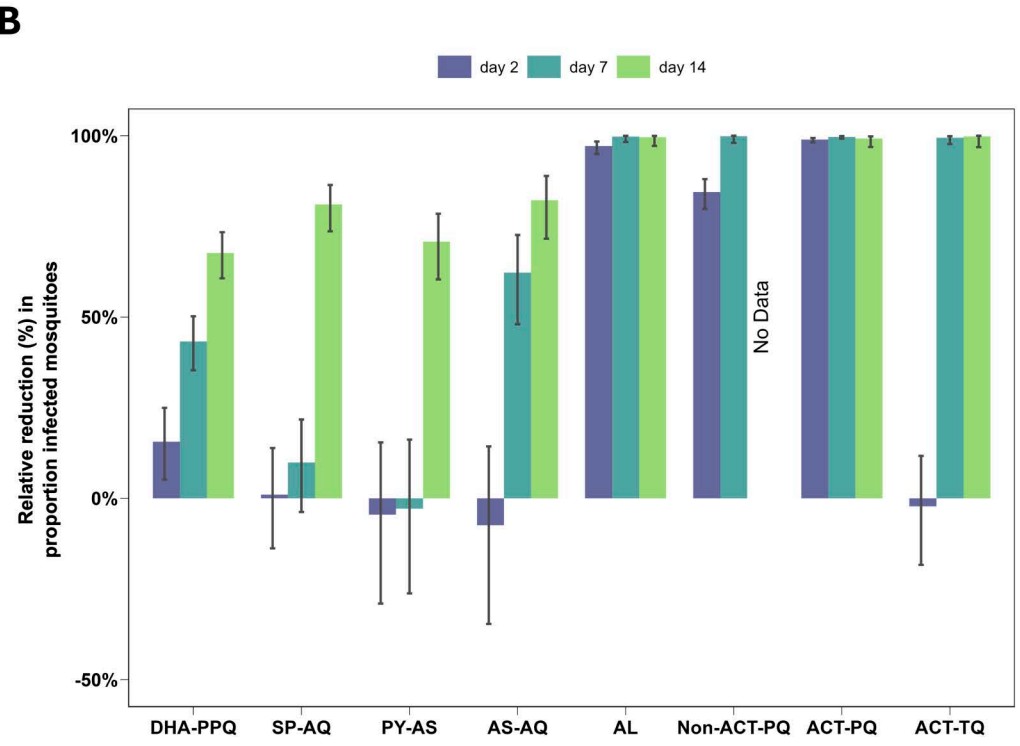

**Fig 4. Infectivity of submicroscopic gametocyte infections and the relative reduction in the proportion of infected mosquitoes compared to baseline per treatment category. A**. Stacked bar chart representing the number of observations for each proportion of infected mosquitoes (rounded to the nearest integer) at baseline and at days 2,7 and 14 post-treatment initiation. For each day of follow-up, individual study participants contribute a single observation. Bars are coloured by the presence of gametocytes by microscopy (blue) or by RT-qPCR only (orange). Baseline visualisations represent a total of 422 study participants; per study 42 (2013-2014), 60 (2016), 100 (2019), 60 (2020), 60 (2021) and 100 (2022) participants were enrolled and presented here. At day 2, data from 375 individuals are presented (60, 99, 60, 58, 98 participants from the 2016, 2019, 2020, 2021 and 2022 studies, respectively). At day 7 post-treatment initiation, 367 participants are presented (56, 98, 59, 58 and 97 participants from the 2016, 2019, 2020, 2021 and 2022 studies, respectively). Finally, at day 14, data from 218 individuals are presented (47, 56, 17 and 96 participants from the 2019, 2020, 2021 and 2022 studies, respectively). Y-axis is log-transformed using log(1 + y) to allow visualisation of zero counts. **B.** Bar chart illustrating the relative reduction compared to baseline in the proportion of infected mosquitoes by treatment group over three time points (Day 2 – indigo, Day 7 – turquoise, and Day 14 – green). Vertical bars depict the 95% confidence intervals for these estimates. Visualisations represent data from 416 individuals at baseline (79, 40, 25, 20, 60, 20, 139 and 40 individuals from the DHA-PPQ, SP-AQ, PY-AS, AS-AQ, AL, non-ACT-PQ, ACT-PQ and ACT-TQ groups, respectively). At day 2, 416 individuals were included (78, 40, 25, 20, 58, 20, 135 and 40 individuals from the DHA-PPQ, SP-AQ, PY-AS, AS-AQ, AL, Non-ACT-PQ, ACT-PQ and ACT-TQ groups, respectively). Data from 409 individuals at day 7 is shown (76, 39, 25, 20, 57, 19, 134 and 39 individuals from the DHA-PPQ, SP-AQ, PY-AS, AS-AQ, AL, Non-ACT-PQ, ACT-PQ and ACT-TQ groups, respectively). Day 14 includes data from 218 individuals (43, 15, 25, 19, 39, 40 and 37 individuals from the DHA-PPQ, SP-AQ, PY-AS, AS-AQ, AL, ACT-PQ and ACT-TQ groups, respectively).

Table L in S1 Appendix, pp 29–32). Relative reductions in oocyst density in dissected mosquitoes preceded the reduction in mosquito infection rates in certain treatment groups, such as DHA-PPQ, SP-AQ, PY-AS and AS-AQ, where a larger reduction in oocyst density was found prior to observing the same level of reduction in mosquito infection rate (Figs O–Q and Table M in S1 Appendix, pp 33–36).

### Clearance of infectivity precedes gametocyte clearance

The clearance of infectivity (i.e., the time until a participant no longer infected any mosquitoes) preceded the clearance of gametocytes assessed by both microscopy and molecular methods (Fig 6 and Table N in S1 Appendix, pp 37–40). Hazard ratios for between-group comparisons and forest plots with between-group comparisons for the clearance of infectivity, adjusted by baseline RT-qPCR gametocyte densities, as well as for gametocyte density measured by microscopy and molecular methods can be found in Tables O–Q and Figs R–T in S1 Appendix (pp 41–46). In the NMA, the treatment groups with SLD PQ exhibited the fastest infectivity clearance, however, this clearance was not significantly slower in the groups treated with AL (hazard ratios 0.89 ($p = 0.8468$) and 0.79 ($p = 0.4357$) compared to non-ACT-PQ and ACT-PQ, respectively). Infectivity was abrogated by day 7 in the AL, ACT-PQ and non-ACT-PQ groups, and by day 14 in the ACT-TQ group, while this effect was slower in the SP-AQ (day 28), DHA-PPQ (day 35) and PY-AS (day 35) groups. Not all infectivity was cleared by the end of follow up (day 28) in the AS-AQ group (4.55% of individuals still infectious, 95% CI 0.67–30.85). ACT-PQ was significantly faster than all other regimens at microscopy-detected gametocyte clearance ($p < 0.0001$ and $p = 0088$ in all comparisons; (Table P in S1 Appendix, p 42). Gametocytes detectable by microscopy persisted after day 28 in the DHA-PPQ, SP-AQ, PY-AS, AL and non-ACT-PQ groups, while gametocytes were microscopically undetectable by day 14 in the ACT-PQ group and by day 21 in the ACT-TQ group. Gametocytes measured by RT-qPCR (limit of detection of 0.01 gametocytes/μL) persisted in a subset of individuals in all treatment groups until the end of follow-up.

### Validity of the network meta-analysis

The NMA assumption of transitivity (see Methods A in S1 Appendix, p 12) was supported by the general similarity in participant characteristics and pre-treatment transmission potential across the included studies. Importantly, for longitudinal outcomes, the primary outcomes were defined as relative reductions from baseline within each individual, which further mitigates concerns related to baseline heterogeneity across studies. These factors support transitivity across the network of treatment comparisons. Visual inspection of the consistency showed that direct and indirect estimates were generally in agreement, with overlapping confidence intervals (Fig U in S1 Appendix, p 47). This suggests that the consistency

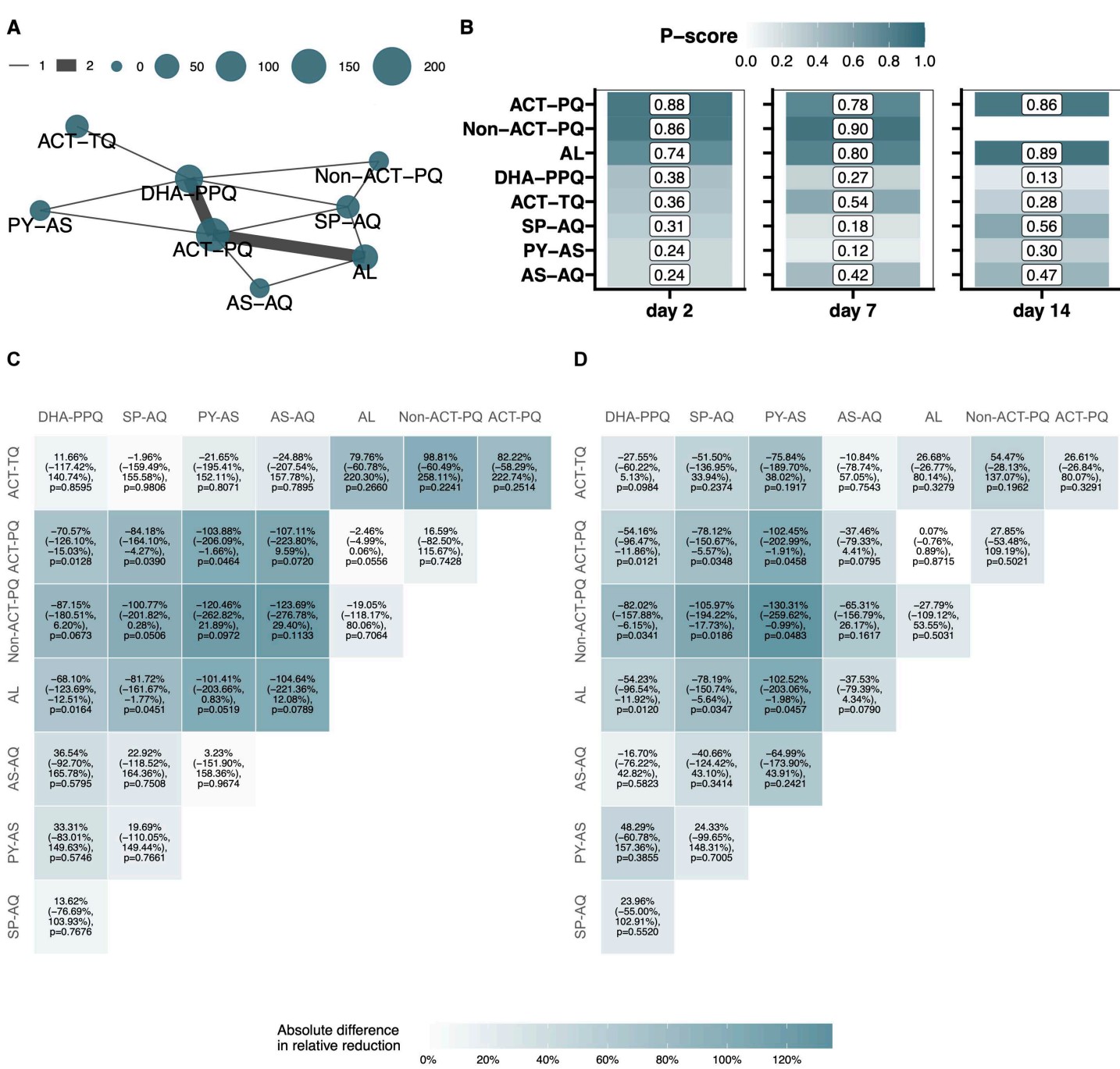

**Fig 5. NMA results and between-group comparison of the reduction in proportion infected mosquitoes at days 2 and 7 post-treatment, compared to baseline. A.** Network plot of treatment comparisons. Node size reflects the total number of individuals receiving each treatment, while the thickness of the connecting lines represents the number of studies comparing the connected treatment pairs. **B.** P-scores, reflecting the relative ranking of treatments (higher scores indicating greater efficacy) for each treatment group on days 2, 7 and 14, ranked by highest p-score on day 2. Values are shown both numerically within each heatmap cell and visually through the colour gradient. **C, D.** Heatmaps representing the absolute difference between treatment groups in the relative reduction in proportion infected mosquitoes at days 2 (**C**) and 7 (**D**) post-treatment, compared to baseline, with 95% CI and p-values. Heatmap cells are coloured by the absolute difference. For example, the absolute difference between DHA-PPQ and AL in the relative reduction in proportion infected mosquitoes at day 2 is −68.10% (−123.69%, −12.51%), and the difference between these groups (68.10% lower reduction for DHA-PPQ) is statistically significant (p = 0.0164). P-values were determined via a two-stage individual participant data network meta-analysis.

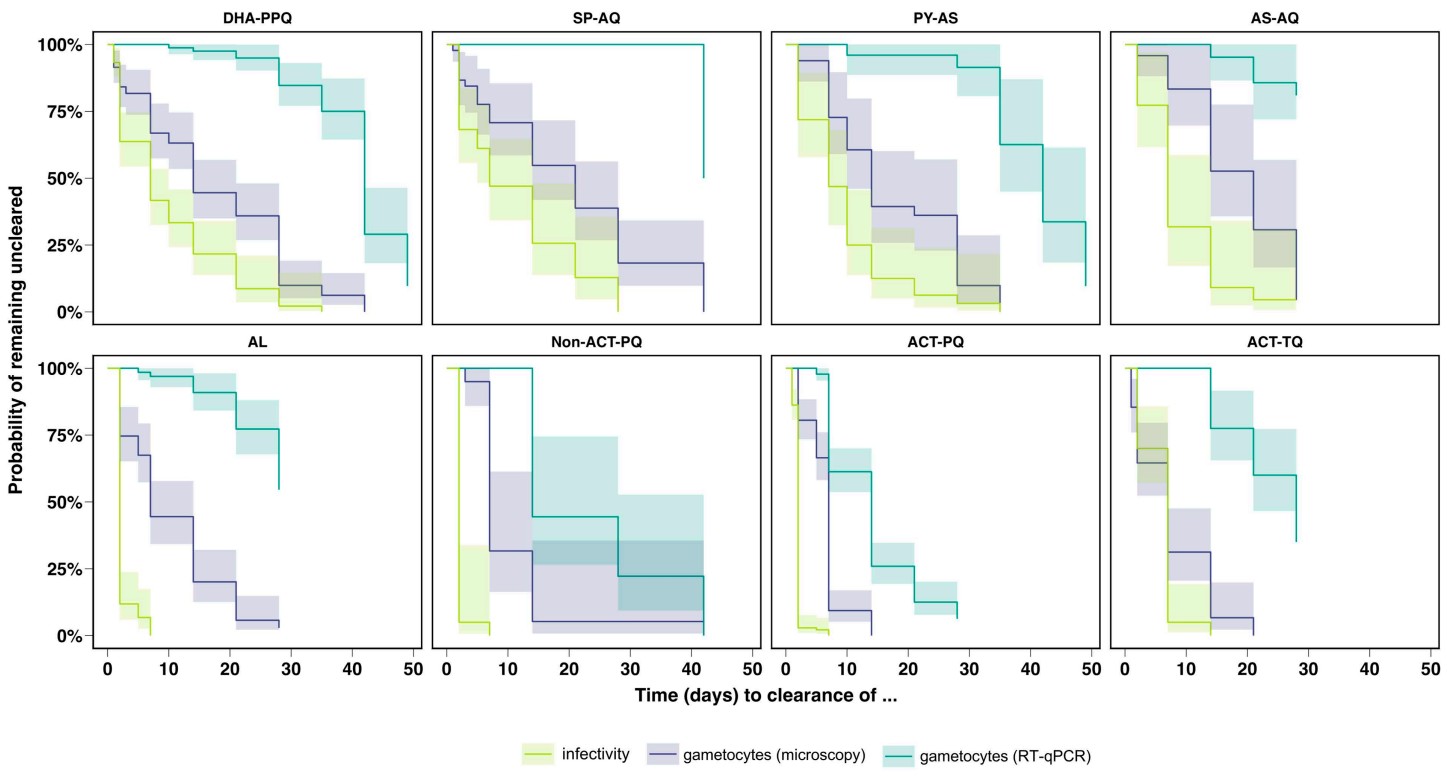

**Fig 6. Time to clearance of infectivity and gametocytes per treatment category.** Kaplan–Meier survival curves showing the cumulative probability of remaining uncleared of gametocytes detected by microscopy (purple), gametocytes detected by RT-qPCR (turquoise) and infectivity to mosquitoes (green) over time stratified across different antimalarial treatment categories (DHA-PPQ, SP-AQ, PY-AS, AS-AQ, AL, Non-ACT-PQ, ACT-PQ, ACT-TQ). Shaded areas indicate 95% confidence intervals for the Kaplan-Meier survival estimates. Survival curves showing infectivity represent data from 417 individuals (79, 40, 25, 20, 58, 20, 135 and 40 individuals from the DHA-PPQ, SP-AQ, PY-AS, AS-AQ, AL, Non-ACT-PQ, ACT-PQ and ACT-TQ groups, respectively). Survival curves visualising gametocytes by microscopy show data from 375 individuals (65, 40, 25, 20, 58, 20, 107 and 40 individuals from the DHA-PPQ, SP-AQ, PY-AS, AS-AQ, AL, Non-ACT-PQ, ACT-PQ and ACT-TQ groups, respectively) and 417 individuals (79, 40, 25, 20, 58, 20, 135 and 40 individuals from the DHA-PPQ, SP-AQ, PY-AS, AS-AQ, AL, Non-ACT-PQ, ACT-PQ and ACT-TQ groups, respectively) for gametocyte assessment by RT-qPCR.

assumption was reasonably upheld across the treatment networks included in this analysis. In general, the network of treatment comparisons was relatively sparse, with most treatment pairs compared in only one or two studies (Fig 5A). This limited connectivity likely contributed to the wide confidence intervals observed for some indirect comparisons, particularly where few or no direct comparisons were available.

## Discussion

This pooled analysis from individual patient data compares the transmission-blocking activity of different regimens of anti-asexual antimalarials with or without a single low-dose gametocytocide in individuals with *P. falciparum* gametocytaemia in Mali. We found a marked difference between the different ACTs, with AL causing the largest reduction in mosquito infection rate (97.15%) within 48 h of treatment initiation. The impact of DHA-PPQ, PY-AS and AS-AQ on transmission was substantially lower than AL and led to prolonged gametocyte carriage and infectivity post-treatment. Adding an SLD PQ (0.25 or 0.5 mg/kg) to any ACT accelerated the clearance of gametocytes and led to a substantially greater reduction in mosquito infection rate within 48 h of treatment for all ACTs except AL.

The effectiveness and utility of the treatments in the six studies analysed here have been discussed in depth in the original trial reports [21–26]. To our knowledge, this combined analysis represents the first synthesis of this data and direct comparison of these treatments. The use of network meta-analysis in this pooled analysis enabled comparisons across a broad set of treatment regimens, including many that were not directly evaluated against each other within individual trials. By estimating reductions from baseline within each treatment group and comparing these reductions across treatment groups, we synthesised both direct and indirect evidence for how the different treatment regimens compare in terms of their gametocytocidal and transmission-blocking activities. While this approach allowed for a more comprehensive assessment of relative efficacy, the network was relatively sparse, with most comparisons informed by only one or two studies. This contributed to wide confidence intervals for some contrasts and added uncertainty to the treatment rankings. These limitations highlight the impact of network sparsity and reinforce the importance of both network structure and statistical precision when interpreting relative treatment rankings. Nonetheless, the consistency observed between direct and indirect estimates, combined with standardised trial procedures, supports the validity of the network-based comparisons.

Prior to these trials, there was sparse evidence of the gametocytocidal and transmission-reducing efficacy for many of the included treatments. The available data frequently originated from single studies, used disparate and insensitive methodologies for gametocyte quantification, or did not conduct transmission assays to determine the transmissibility of gametocytes [37–41]. This was the rationale for including only studies with highly similar enrolment criteria and assessments, including transmission assays before and after initiation of treatment. This pooled analysis confirms SP-AQ's poor ability to clear gametocytes and reduce transmission, with a substantially lower relative reduction in mosquito infection rate at day 7 compared to DHA-PPQ, AS-AQ, and AL, although this difference was not significant for ACTs other than AL. As the only non-artemisinin-based antimalarial recommended for systematic distribution via seasonal malaria chemoprevention (SMC) [42], SP-AQ's poor performance against gametocytes may affect the overall community impact of SMC. It remains to be determined whether adding a single low dose of primaquine to SP-AQ for chemoprevention would lead to substantial community benefits. AL led to substantially larger reductions in mosquito infection rate on day 2 after initiation of treatment compared to DHA-PPQ, PY-AS, SP-AQ and AS-AQ. Whilst superior gametocyte-clearing effects of AL have previously been demonstrated when compared with DHA-PPQ [43] and PY-AS [44], the reason is not entirely clear since lumefantrine only exhibits an effect on gametocytes and transmission at concentrations higher than therapeutic levels [2,5–7]. The total dose of the artemisinin component may be higher in AL compared to DHA-PPQ. In addition, lumefantrine may exert an effect on oocysts by impairing sporogony without significantly affecting male gamete exflagellation [45]. Similar observations have been reported in another study, which demonstrated a reduction in oocyst numbers at the lumefantrine IC50 concentration, while effects on gametocytes were only evident at concentrations five times the IC50 [2].

To our knowledge, this pooled analysis provides the first direct comparison of SLD PQ and SLD TQ with the same partner drug. Our analysis indicates a delayed but highly effective response of SLD TQ compared to SLD PQ when combined with ACTs, achieving a 99.42% reduction in mosquito infection rate at day 7 from baseline. Although the transmission-blocking efficacy of TQ is dose-dependent, our pooled analysis combined the two highest dose groups (0.83 mg/kg and 1.66 mg/kg) for simplicity, possibly obscuring the effects of the higher dose. Despite the delayed transmission-blocking properties of TQ compared with SLD PQ in our findings, its long half-life could be a major advantage and could be of relevance to prevent the transmission of drug-resistant parasites, which may have an increased gametocyte conversion rate and transmission potential [46,47]. It is important to note that TQ was evaluated only in combination with a single ACT, DHA-PPQ, in these studies, whereas PQ was assessed in combination with multiple ACTs, including DHA-PPQ, AL, PY-AS, and AS-AQ.

Before ACTs were introduced, several antimalarials, including chloroquine and sulfadoxine-pyrimethamine were reported to increase gametocytaemia and/or infectivity after treatment [48,49]. Our analysis, almost exclusively examining ACTs, could not rule out a certain degree of post-treatment enhancement of infectivity. Although enhancements observed in our analyses were small and transient, we did observe that a non-negligible proportion of individuals became infectious

after treatment (this proportion being ≥10% of participants in the DHA-PPQ, SP-AQ, PY-AS and AS-AQ groups); this percentage was below 2% in the other treatment groups, including AL. The fact that very few individuals become gametocyte positive or infectious after AL suggests that, among ACT regimens, AL may be superior at targeting immature or sequestered gametocytes.

The presence of gametocytes after treatment at densities that normally permit transmission does not make transmission an inevitability, because gametocytes may be sterilised or become non-infectious for other reasons before being cleared [12]. In this pooled analysis, this discordance was most pronounced in the AL group and groups with SLD PQ, where post-treatment effects on infectivity preceded gametocyte clearance. In all other treatment groups, reductions in gametocyte density and mosquito infection rate followed more similar patterns. The combination of an ACT with SLD PQ was significantly faster than all other regimens at gametocyte clearance (100% of microscopy-detected gametocytes by day 14), while ACT-TQ was the second fastest regimen to clear gametocytes (100% of microscopy-detected gametocytes by day 21). Infectivity clearance was the fastest in the treatment groups with SLD PQ, followed by AL.

A few limitations in our study warrant consideration. Firstly, we established transmission-blocking activity in highly infectious individuals. This population allows a detailed assessment of transmission-blocking properties; however, post-treatment transmission potential is likely to be smaller in the majority of malaria patients who predominantly have sub-microscopic gametocyte densities or may even be free of gametocytes at clinical presentation. A different study would be needed to determine the relative importance of submicroscopic versus microscopy-detected gametocyte carriage for malaria transmission in clinical patients though these would likely require larger sample sizes for mosquito infectivity endpoints. Secondly, this analysis reuses data from studies that were highly similar in enrolment criteria and methodology but not originally designed for the conduction of a pooled analysis. However, a number of treatments were tested in multiple studies (DHA-PPQ, DHA-PPQ plus 0.25 mg/kg PQ, AL and SP-AQ) and our results were highly concordant across years (Fig K in S1 Appendix, p. 28). We interpret this as evidence of the value of established transmission testing facilities and the appropriateness of a pooled analysis. Thirdly, certain treatment groups included in the analysis had small sample sizes, resulting in wide confidence intervals in the relative reductions and limiting the ability to draw robust conclusions for these groups. The sample size in individual studies were defined to detect large (>90%) reductions in transmission to mosquitoes [50], and may have been insufficient to detect more modest transmission reductions. Our NMA was also challenged by the limited number of direct comparisons of study groups that resulted in wide confidence intervals and uncertainty in some comparisons and rankings. For instance, although ACT-TQ showed large reductions from baseline across several outcomes, its P-scores were lower than expected (Fig 5B). This likely reflects the fact that ACT-TQ was included in only a single study and had limited overlap with other treatment groups, which meant that comparisons involving ACT-TQ relied heavily on indirect evidence with large standard errors.

Furthermore, the studies included in this analysis were conducted over the course of a decade, and susceptibility of parasites to antimalarials may have changed over time. Moreover, our study utilised direct membrane feeding assays rather than direct skin feeding, which may have resulted in reduced infectivity, despite no compelling evidence of gametocytes sequestering in the skin [51]. In addition, we measured oocyst prevalence instead of sporozoite prevalence and some infected mosquitoes may not have become infectious [52]. Nevertheless, comparisons between groups remain valid, and we observed a strong association between oocyst prevalence and oocyst densities (Fig B in S1 Appendix, p. 9), indicating that a higher proportion of infected mosquitoes reflects a greater transmission potential. While reductions in transmission potential among treated individuals are encouraging, they may not directly translate to reductions in overall malaria transmission in communities and this depends on the populations in which antimalarials are deployed. Asymptomatic and untreated infections likely contribute substantially to ongoing transmission in many settings. Such infections may be targeted through mass drug administration with non-artemisinin-based treatments such as SP-AQ, for example, in the context of SMC; our individual-level data suggest that adding a gametocytocide to this combination could provide additional benefits. Lastly, this pooled analysis set out to assess the ability of antimalarial treatment to prevent

transmission. However, considering that gametocytes in *PfKelch13* mutant infections might preferentially survive artemisinin exposure and infect mosquitoes [47], our findings regarding ACTs may not be generalisable to areas where artemisinin partial resistance is present. In addition, increased gametocytaemia was found to be an early indicator of resistance emergence against previous non-ACT first-line treatments [46]. Consequently, while our results indicate that AL is nearly as effective as ACT-PQ in reducing mosquito infection rates, it remains unclear whether this would extend to settings with artemisinin-resistant malaria infections. Our data on SLD gametocytocides in combination with ACTs therefore supports the 2023 advice from WHO malaria policy and advisory group to expand the focus on reducing parasite transmission with SLD PQ in areas where partial artemisinin resistance has been detected [20]. Of note, evidence is needed to support this expert advice on the utility of SLD PQ. Moreover, resistance to artemisinin partners drugs is of particular importance for ACT efficacy; there are concerns that susceptibility of parasites to lumefantrine is decreasing in some settings in Africa [53]. While we observed high transmission-blocking efficacy of AL when administered as a directly observed therapy; its efficacy may be lower in real-life settings where treatment adherence may be lower.

In conclusion, utilising individual patient-level data from six clinical trials conducted at the same study site with highly consistent transmission experiments across trials, we showed pronounced differences in anti-gametocyte and anti-transmission effects between ACTs, with AL showing highest efficacy in blocking transmission. Additionally, our findings emphasise the rapid effects of SLD PQ in clearing and sterilising gametocytes when used in combination with an ACT, while the addition of SLD TQ to ACTs has a delayed transmission-blocking effect compared to SLD PQ. Lastly, our analysis confirms considerable post-treatment transmission after SP-AQ, indicating that adding SLD PQ to SP-AQ may be considered to more effectively block transmission in community treatment campaigns.

## Supporting information

**S1 Appendix.** **Text A. Antimalarial treatment dosage.** Dosing schedules for each treatment regimen. **Table A. Antimalarial treatment suppliers.** Suppliers of study drugs used in each study. **Fig A. Original PQ03 molecular gametocyte densities.** Bland-Altman plot presenting the agreement between gametocyte density measured by microscopy (x-axis) and RT-qPCR (y-axis) pre-treatment. The original PQ03 (2016) RT-qPCR data is presented here, which showed that the comparison of densities measured by both methods for this study was an outlier compared to the other studies. The solid black line indicates the overall mean difference including all studies, and dotted lines represent 1.96 x standard deviation of the differences. The dashed lines, coloured by study, represent group-specific mean differences. Only in 2016 (PQ03 study), there was a relevant difference between the two gametocyte measurements; molecular assays were repeated for this sample set prior to inclusion in the analyses. **Table B. Baseline descriptives per treatment group.** Study participants characteristics, parasite densities and infectivity prior to treatment. **Fig B. Relation between oocyst density and prevalence.** Association between oocyst prevalence and oocyst density, determined by a mixed-effects logistic regression with random effect for study, where the solid black line represents the association averaged over studies. **Table C. Relative reduction in gametocyte prevalence.** Relative reduction compared to baseline in gametocyte prevalence by microscopy and RT-qPCR, at three time points (Day 2, Day 7, Day 14), with 95% confidence intervals. **Fig C. Relative reduction in gametocyte prevalence per study arm (ungrouped).** Bar charts illustrating the relative reduction compared to baseline in gametocyte prevalence by microscopy and RT-qPCR for each study arm (ungrouped), over three time points (Day 2, Day 7, Day 14). Vertical bars depict the 95% confidence intervals for these estimates. **Methods A**. **Network meta-analysis (NMA) methodology and assessment of validity.** This section outlines the NMA, which integrated direct and indirect comparisons across studies to estimate relative treatment effects. It details the assessment of key NMA assumptions, transitivity and consistency, including the comparability of participant characteristics and outcomes across studies, and the agreement between direct and indirect treatment effects. **Fig D. Forest plots of treatment comparisons of reduction in gametocyte prevalence by microscopy at days 2, 7, and 14.** All comparisons are with ACT-PQ. Results from the network meta-analysis are shown as mean

differences (MD) in relative reductions from baseline in gametocyte prevalence by microscopy, with 95% confidence intervals. Each treatment is compared to ACT-PQ, the reference treatment. Negative values indicate a smaller reduction than ACT-PQ, while positive values indicate a larger reduction. Point estimates are plotted as squares proportional to study weight, and horizontal lines denote confidence intervals. **Fig E. Forest plots of treatment comparisons of reduction in gametocyte prevalence by RT-qPCR at days 2, 7, and 14**. All comparisons are with ACT-PQ. Results from the network meta-analysis are shown as mean differences (MD) in relative reductions from baseline in gametocyte prevalence by RT-qPCR, with 95% confidence intervals. Each treatment is compared to ACT-PQ, the reference treatment. Negative values indicate a smaller reduction than ACT-PQ, while positive values indicate a larger reduction. Point estimates are plotted as squares proportional to study weight, and horizontal lines denote confidence intervals. **Table D. Relative reduction in gametocyte density.** Relative reduction compared to baseline in gametocyte density by microscopy and RT-qPCR, at three time points (Day 2, Day 7, Day 14), with 95% confidence intervals. **Fig F. Relative reduction in gametocyte density per study arm (ungrouped).** Bar charts illustrating the relative reduction compared to baseline in gametocyte prevalence by microscopy and RT-qPCR for each study arm (ungrouped), over three time points (Day 2, Day 7, Day 14). Vertical bars depict the 95% confidence intervals for these estimates. **Table E. Treatment comparisons of reduction in gametocyte density by microscopy at day 2.** Results from the network meta-analysis are shown as the absolute difference between treatment groups in the relative reduction from baseline in gametocyte density by microscopy at day 2, with 95% confidence intervals and corresponding $p$-values. For example, the absolute difference between DHA-PPQ and AL in the relative reduction in microscopical gametocyte density at day 2 is −19.56% (−97.38%, 58.26%), and the difference between these arms (19.56% lower reduction for DHA-PPQ) is not statistically significant ($p = 0.6223$). **Table F. Treatment comparisons of reduction in gametocyte density by microscopy at day 7.** Results from the network meta-analysis are shown as the absolute difference between treatment groups in the relative reduction from baseline in gametocyte density by microscopy at day 7, with 95% confidence intervals and corresponding $p$-values. For example, the absolute difference between DHA-PPQ and AL in the relative reduction in microscopical gametocyte density at day 7 is −47.87% (−94.78%, −0.96%), and the difference between these arms (47.87% lower reduction for DHA-PPQ) is statistically significant ($p = 0.0455$). **Table G. Treatment comparisons of reduction in gametocyte density by microscopy at day 14.** Results from the network meta-analysis are shown as the absolute difference between treatment groups in the relative reduction from baseline in gametocyte density by microscopy at day 14, with 95% confidence intervals and corresponding $p$-values. For example, the absolute difference between DHA-PPQ and AL in the relative reduction in microscopical gametocyte density at day 14 is −7.58% (−22.68%, 7.53%), and the difference between these arms (7.58% lower reduction for DHA-PPQ) is not statistically significant ($p = 0.3255$). **Table H. Treatment comparisons of reduction in gametocyte density by RT-qPCR at day 2.** Results from the network meta-analysis are shown as the absolute difference between treatment groups in the relative reduction from baseline in gametocyte density by RT-qPCR at day 2, with 95% confidence intervals and corresponding $p$-values. For example, the absolute difference between DHA-PPQ and AL in the relative reduction in molecular gametocyte density at day 2 is −12.51% (−81.35%, 56.34%), and the difference between these arms (12.51% lower reduction for DHA-PPQ) is not statistically significant ($p = 0.7218$). **Table I. Treatment comparisons of reduction in gametocyte density by RT-qPCR at day 7.** Results from the network meta-analysis are shown as the absolute difference between treatment groups in the relative reduction from baseline in gametocyte density by RT-qPCR at day 7, with 95% confidence intervals and corresponding $p$-values. For example, the absolute difference between DHA-PPQ and AL in the relative reduction in molecular gametocyte density at day 7 is −18.50% (−54.50%, 17.49%), and the difference between these arms (18.50% lower reduction for DHA-PPQ) is not statistically significant ($p = 0.3137$). **Table J. Treatment comparisons of reduction in gametocyte density by RT-qPCR at day 14.** Results from the network meta-analysis are shown as the absolute difference between treatment groups in the relative reduction from baseline in gametocyte density by RT-qPCR at day 14, with 95% confidence intervals and corresponding $p$-values. For example, the absolute difference between DHA-PPQ

and AL in the relative reduction in molecular gametocyte density at day 14 is −19.35% (−39.05%, 0.36%), and the difference between these arms (19.35% lower reduction for DHA-PPQ) is nearly statistically significant ($p = 0.0543$). **Fig G. Forest plots of treatment comparisons of reduction in gametocyte density by microscopy at days 2, 7, and 14.** All comparisons are with ACT-PQ. Results from the network meta-analysis are shown as mean differences (MD) in relative reductions from baseline in gametocyte density by microscopy, with 95% confidence intervals. Each treatment is compared to ACT-PQ, the reference treatment. Negative values indicate a smaller reduction than ACT-PQ, while positive values indicate a larger reduction. Point estimates are plotted as squares proportional to study weight, and horizontal lines denote confidence intervals. **Fig H. Forest plots of treatment comparisons of reduction in gametocyte density by RT-qPCR at days 2, 7, and 14.** All comparisons are with ACT-PQ. Results from the network meta-analysis are shown as mean differences (MD) in relative reductions from baseline in gametocyte density by RT-qPCR, with 95% confidence intervals. Each treatment is compared to ACT-PQ, the reference treatment. Negative values indicate a smaller reduction than ACT-PQ, while positive values indicate a larger reduction. Point estimates are plotted as squares proportional to study weight, and horizontal lines denote confidence intervals. **Table K. Relative reduction in proportion infected mosquitoes.** Relative reduction compared to baseline in proportion infected mosquitoes, at three time points (Day 2, Day 7, Day 14), with 95% confidence intervals and corresponding *p*-values. **Fig I. Forest plots of treatment comparisons of reduction in proportion infected mosquitoes at days 2, 7, and 14.** All comparisons are with ACT-PQ. Results from the network meta-analysis are shown as mean differences (MD) in relative reductions from baseline in proportion infected mosquitoes, with 95% confidence intervals. Each treatment is compared to ACT-PQ, the reference treatment. Negative values indicate a smaller reduction than ACT-PQ, while positive values indicate a larger reduction. Point estimates are plotted as squares proportional to study weight, and horizontal lines denote confidence intervals. **Fig J. Relative reduction in proportion infected mosquitoes per study arm (ungrouped).** Bar charts illustrating the relative reduction compared to baseline in the proportion infected mosquitoes for each study arm (ungrouped), over three time points (Day 2, Day 7, Day 14). Vertical bars depict the 95% confidence intervals for these estimates. **Fig K. Relative reduction in proportion infected mosquitoes comparing the same study arms across different studies.** Bar charts illustrating the relative reduction compared to baseline in the proportion infected mosquitoes for each study arm over three time points (Day 2, Day 7, Day 14) and for each study that it was evaluated in. Vertical bars depict the 95% confidence intervals for these estimates. **Fig L. Relative reduction in the probability of infecting at least 1 mosquito.** Relative reduction compared to baseline in the probability of infecting at least 1 mosquito, at three time points (Day 2, Day 7, Day 14), with 95% confidence intervals. **Table L. Relative reduction in the probability of infecting at least 1 mosquito.** Relative reduction compared to baseline in the probability of infecting at least 1 mosquito, at three time points (Day 2, Day 7, Day 14), with 95% confidence intervals and corresponding *p*-values. **Fig M. Relative reduction in the probability of infecting at least 1 mosquito per study arm (ungrouped).** Bar charts illustrating the relative reduction compared to baseline in the probability of infecting at least 1 mosquito for each study arm (ungrouped), over three time points (Day 2, Day 7, Day 14). Vertical bars depict the 95% confidence intervals for these estimates. **Fig N. Forest plots of treatment comparisons of probability of infecting at least 1 mosquito at days 2, 7, and 14.** All comparisons are with ACT-PQ. Results from the network meta-analysis are shown as mean differences (MD) in relative reductions from baseline in infectiousness, with 95% confidence intervals. Each treatment is compared to ACT-PQ, the reference treatment. Negative values indicate a smaller reduction than ACT-PQ, while positive values indicate a larger reduction. Point estimates are plotted as squares proportional to study weight, and horizontal lines denote confidence intervals. **Fig O. Relative reduction in oocyst density.** Relative reduction compared to baseline in oocyst density, at three time points (Day 2, Day 7, Day 14), with 95% confidence intervals. **Table M. Relative reduction in oocyst density.** Relative reduction compared to baseline in oocyst density, at three time points (Day 2, Day 7, Day 14), with 95% confidence intervals and corresponding *p*-values. **Fig P. Relative reduction in oocyst density per study arm (ungrouped).** Bar charts illustrating the relative reduction compared to baseline in oocyst density for each study arm (ungrouped), over

three time points (Day 2, Day 7, Day 14). Vertical bars depict the 95% confidence intervals for these estimates. **Fig Q. Forest plots of treatment comparisons of reduction in oocyst density at days 2, 7, and 14.** All comparisons are with ACT-PQ. Results from the network meta-analysis are shown as mean differences (MD) in relative reductions from baseline in oocyst density, with 95% confidence intervals. Each treatment is compared to ACT-PQ, the reference treatment. Negative values indicate a smaller reduction than ACT-PQ, while positive values indicate a larger reduction. Point estimates are plotted as squares proportional to study weight, and horizontal lines denote confidence intervals. **Table N. Kaplan–Meier survival data for time to clearance of infectivity and gametocytes, by treatment arm.** Data from Kaplan-Meier survival curves showing the cumulative probability of remaining uncleared of gametocytes detected by microscopy, gametocytes detected by RT-qPCR and infectivity to mosquitos over time stratified across different antimalarial treatment categories (DHA-PPQ, SP-AQ, PY-AS, AS-AQ, AL, Non-ACT-PQ, ACT-PQ, ACT-TQ). The number of individuals at risk per time point is presented (n.risk), as well as the number of clearance events (n.event) and the number of censored observations (n.censor). **Table O. Hazard ratios for infectivity survival curves (adjusted by baseline PCR gametocyte densities).** Hazard ratios for between-arm comparisons of infectivity clearance, with corresponding 95% CI and *p*-value. For example, hazard ratio of infectivity clearance for AL compared to DHA-PPQ is 6.12, meaning that clearance is 6.12 times more likely to take place in the AL group compared to the DHA-PPQ group, and this is significantly different ($p < 0.0001$). [a]Indirect comparison in the network meta-analysis. **Table P. Hazard ratios for gametocytes by microscopy survival curves.** Hazard ratios for between-arm comparisons of microscopical gametocyte clearance, with corresponding 95% CI and *p*-value. For example, hazard ratio of microscopical gametocyte clearance for AL compared to DHA-PPQ is 2.60, meaning that clearance is 2.60 times more likely to take place in the AL group compared to the DHA-PPQ group, and this is significantly different ($p = 0.0012$). [a]Indirect comparison in the network meta-analysis. **Table Q. Hazard ratios for gametocytes by RT-qPCR survival curves.** Hazard ratios for between-arm comparisons of molecular gametocyte clearance, with corresponding 95% CI and *p*-value. For example, hazard ratio of molecular gametocyte clearance for AL compared to DHA-PPQ is 0.92, meaning that clearance is 0.92 times more likely to take place in the AL group compared to the DHA-PPQ group, and this is significantly different ($p = 0.8575$). [a]Indirect comparison in the network meta-analysis. **Fig R. Forest plots of treatment comparisons of infectivity survival curves (adjusted by baseline RT-qPCR gametocyte densities) at days 2, 7, and 14.** All comparisons are with ACT-PQ. Results from the network meta-analysis are shown as mean differences (MD) in relative reductions from baseline in infectivity survival curve, with 95% confidence intervals. Each treatment is compared to ACT-PQ, the reference treatment. Negative values indicate a smaller reduction than ACT-PQ, while positive values indicate a larger reduction. Point estimates are plotted as squares proportional to study weight, and horizontal lines denote confidence intervals. **Fig S. Forest plots of treatment comparisons of microscopy gametocyte density survival curves at days 2, 7, and 14.** All comparisons are with ACT-PQ. Results from the network meta-analysis are shown as mean differences (MD) in relative reductions from baseline in gametocyte density survival curve, with 95% confidence intervals. Each treatment is compared to ACT-PQ, the reference treatment. Negative values indicate a smaller reduction than ACT-PQ, while positive values indicate a larger reduction. Point estimates are plotted as squares proportional to study weight, and horizontal lines denote confidence intervals. **Fig T. Forest plots of treatment comparisons of molecular gametocyte density survival curves at days 2, 7, and 14.** All comparisons are with ACT-PQ. Results from the network meta-analysis are shown as mean differences (MD) in relative reductions from baseline in molecular gametocyte density survival curve, with 95% confidence intervals. Each treatment is compared to ACT-PQ, the reference treatment. Negative values indicate a smaller reduction than ACT-PQ, while positive values indicate a larger reduction. Point estimates are plotted as squares proportional to study weight, and horizontal lines denote confidence intervals. **Fig U. Consistency assessment comparing direct and indirect treatment effects for proportion infected mosquitoes.** Consistency assessment comparing direct (blue) and indirect (red) treatment effects for proportion infected mosquitoes. For each pairwise comparison, the direct effect estimate (based on head-to-head trials) and the indirect effect estimate (derived

from the network via a common comparator) are plotted with their corresponding 95% confidence intervals. Overlapping intervals suggest consistency between direct and indirect evidence, supporting the validity of the network meta-analysis assumptions.
(DOCX)

**S2 Appendix. Summary of studies included in the pooled analysis.**
(XLSX)

**S1 PRISMA Checklist. PRISMA reporting checklist for NMA.**
(DOCX)

## Acknowledgments

We would like to thank everyone involved in the original studies, including the local safety monitor, members of the data safety and monitoring board, and all MRTC study staff for their assistance and support. Finally, we are thankful to all study participants and the population of Ouélessébougou, Mali, for their cooperation.

## Author contributions

**Conceptualisation:** Leen N. Vanheer, Jordache Ramjith, Almahamoudou Mahamar, Roly Gosling, Joelle M. Brown, Chris Drakeley, Alassane Dicko, Will Stone, Teun Bousema.

**Data curation:** Leen N. Vanheer, Jordache Ramjith, Almahamoudou Mahamar, Merel J. Smit, Michelle E. Roh, Will Stone.

**Formal analysis:** Leen N. Vanheer, Jordache Ramjith.

**Funding acquisition:** Roly Gosling, Joelle M. Brown, Chris Drakeley, Alassane Dicko, Will Stone, Teun Bousema.

**Investigation:** Leen N. Vanheer, Almahamoudou Mahamar, Merel J. Smit, Michelle E. Roh, Koualy Sanogo, Youssouf Sinaba, Sidi M. Niambele, Makonon Diallo, Seydina O. Maguiraga, Sekouba Keita, Siaka Samake, Ahamadou Youssouf, Halimatou Diawara, Sekou F. Traore, Roly Gosling, Joelle M. Brown, Chris Drakeley, Alassane Dicko, Will Stone, Teun Bousema.

**Methodology:** Almahamoudou Mahamar, Kjerstin Lanke, Koualy Sanogo, Youssouf Sinaba, Sidi M. Niambele, Makonon Diallo, Seydina O. Maguiraga, Sekouba Keita, Siaka Samake, Ahamadou Youssouf, Halimatou Diawara, Sekou F. Traore, Alassane Dicko, Will Stone, Teun Bousema.

**Supervision:** Roly Gosling, Joelle M Brown, Chris Drakeley, Alassane Dicko, Will Stone, Teun Bousema.

**Validation:** Leen N. Vanheer, Jordache Ramjith, Will Stone.

**Visualisation:** Leen N. Vanheer, Jordache Ramjith.

**Writing – original draft:** Leen N. Vanheer, Jordache Ramjith, Joelle M. Brown, Chris Drakeley, Will Stone, Teun Bousema.

**Writing – review & editing:** Leen N. Vanheer, Jordache Ramjith, Almahamoudou Mahamar, Merel J. Smit, Roly Gosling, Joelle M. Brown, Chris Drakeley, Alassane Dicko, Will Stone, Teun Bousema.

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
