## [Editor Report · Decision Letter 0]

25 Feb 2025

Dear Dr Bousema,

Thank you for submitting your manuscript entitled "The transmission blocking activity of artemisinin-combination, non-artemisinin, and 8-aminoquinoline antimalarial therapies: a pooled analysis of individual participant data." for consideration by PLOS Medicine.

Your manuscript has now been evaluated by the PLOS Medicine editorial staff as well as by an academic editor with relevant expertise and I am writing to let you know that we would like to send your submission out for external peer review. Please note that we will return the manuscript to 2 of the original referees.

Please re-submit your manuscript within two working days, i.e. by Feb 27 2025 11:59PM.

Please email me upon submission of the manuscript (afarrell@plos.org) to alert me of its progress.

Kind regards,

Alison

Alison Farrell, Ph.D.

Senior Editor

PLOS Medicine

---

## [Decision Letter · Decision Letter 1]

21 Apr 2025

Dear Dr Bousema,

Many thanks for submitting your manuscript "The transmission blocking activity of artemisinin-combination, non-artemisinin, and 8-aminoquinoline antimalarial therapies: a pooled analysis of individual participant data." (PMEDICINE-D-25-00573R1) to PLOS Medicine. The paper has been reviewed by subject experts and a statistician; their comments are included below and can also be accessed here: [LINK]

As you will see, the statistical reviewer requires that you include additional information to clarify certain aspects of your analyses, including a complete reporting of the NMA in the results, and suggests alternate methods of presentation in some instances. I am also appending comments from the academic editor who requests additional and transparent discussion of the study's limitations. After discussing the paper with the editorial team and the academic editor, I'm pleased to invite you to revise the paper in response to the reviewers' comments. We plan to send the revised paper to some or all of the original reviewers, and we cannot provide any guarantees at this stage regarding publication.

We ask that you submit your revision by May 01 2025 11:59PM. However, if this deadline is not feasible, please contact me by email, and we can discuss a suitable alternative.

Don't hesitate to contact me directly with any questions (afarrell@plos.org).

Best regards,

Alison

Alison Farrell, Ph.D.

Senior Editor

PLOS Medicine

afarrell@plos.org

Summarized comments from the academic editor:

1. The studies included in this analysis were conducted for purposes other than future metanalysis. If a comparison of all study arms were intended the design of the studies would have been different. The authors are reusing study data for a purpose other than answering the original research questions. Please acknowledge this inherent limitation of data re-use.

2. In general, the reporting of the study limitations should be expanded. For example, the enrollment in some studies is a decade apart. Considering that the susceptibility of parasites to antimalarials is changing over time, and changes in the susceptibility of gametocytes precedes other markers of resistance, the authors are not comparing same with same. This should be acknowledged.

3. A reduction in infectivity is not the same as a reduction in transmission. To my knowledge no empirical evidence that such a reduction in infectivity equals a reduction in malaria transmission. It is not difficult to imagine a situation in which most members of a population carry subclinical infections and continue to infect mosquitoes thus contributing to transmission. Individuals treated for clinical malaria make up only a small fraction of the overall population. Treating such a small sub-population will probably make no meaningful difference in the overall transmission prevalence of infections, morbidity, or mortality. This should be acknowledged.

4. The authors state, ”Our data on SLD gametocytocides in combination with ACTs therefore supports the 2023 advice from the WHO malaria policy and advisory group to expand the focus on reducing parasite transmission with SLD PQ in areas where partial artemisinin resistance has been detected (20).” The authors need to acknowledge the need to support the expert advice with evidence. Moreover, if one or more of the authors were part of the WHO expert group that provided this advice, or participated in a WHO malaria policy and advisory group, the authors should declare potential for conflicting interests to ensure transparency and accountability.

Comments from the reviewers:

Reviewer #1: This is an excellent revision by Dr. Bousema and colleagues. I was the first reviewer and had quite detailed comments that were fully addressed. In response to the statistical reviewer, the authors now include a network meta-analysis that has helped strengthen the statistical analysis while not changing core findings from the comparative analysis across the six clinical data sets (five of which have been published as peer-reviewed articles with the sixth being on a preprint archive).

The study contains a wealth of important information about the effects of the different artemisinin-based combination therapies (ACTs) on Plasmodium falciparum transmissibility. The findings of superior transmission-blocking activity of AL (artemether-lumefantrine) compared with other ACTs, and the analysis of transmission-blocking by single low dose 8-aminoaquinolines, are very valuable in the context of current discussions about the need to diversity ACTs across Africa to reduce selective pressure on lumefantrine. This is especially relevant given the increasing number of African countries where artemisinin partial resistance has been identified.

Overall, I find this to be a very high quality and timely addition to the literature that will generate considerable interest in the malaria research and treatment and control communities.

Reviewer #2: See attachment

Michael Dewey

---

* Please upload any figures associated with your paper as individual TIF or EPS files with 300dpi resolution at resubmission; please read our figure guidelines for more information on our requirements: http://journals.plos.org/plosmedicine/s/figures. While revising your submission, please upload your figure files to the PACE digital diagnostic tool, https://pacev2.apexcovantage.com/. PACE helps ensure that figures meet PLOS requirements. To use PACE, you must first register as a user. Then, login and navigate to the UPLOAD tab, where you will find detailed instructions on how to use the tool. If you encounter any issues or have any questions when using PACE, please email us at PLOSMedicine@plos.org.

* Please ensure that the study is reported according to the appropriate guideline and include the relevant completed checklist as Supporting Information. When completing the checklist, please use section and paragraph numbers, rather than page numbers. Please add the following statement, or similar, to the Methods: "This study is reported as per [XXXX] guideline (S1 Checklist)."

FIGURES AND TABLES

SUPPLEMENTARY MATERIAL

REFERENCES

* For all observational studies, in the manuscript text, please indicate: (1) the specific hypotheses you intended to test, (2) the analytical methods by which you planned to test them, (3) the analyses you actually performed, and (4) when reported analyses differ from those that were planned, transparent explanations for differences that affect the reliability of the study's results. If a reported analysis was performed based on an interesting but unanticipated pattern in the data, please be clear that the analysis was data driven.

* Please state in the Methods section whether the study had a prospective protocol or analysis plan. If a prospective analysis plan (from your funding proposal, IRB or other ethics committee submission, study protocol, or other planning document written before analyzing the data) was used in designing the study, please include the relevant document(s) with your revised manuscript as a Supporting Information file to be published alongside your study and cite it in the Methods section. A legend for this file should be included at the end of your manuscript. If no such document exists, please make sure that the Methods section transparently describes when analyses were planned, and when/why any data-driven changes to analyses took place. Changes in the analysis, including those made in response to peer review comments, should be identified as such in the Methods section of the paper, with rationale.

MODELLING STUDIES

The following list is derived from Geoffrey P Garnett, Simon Cousens, Timothy B Hallett, Richard Steketee, Neff Walker. Mathematical models in the evaluation of health programmes. (2011) Lancet DOI:10.1016/S0140-6736(10)61505-X:

* If pertinent, please provide a diagram that shows the model structure, including how the natural history of the disease is represented, the process and determinants of disease acquisition, and how the putative intervention could affect the system.

* Please provide a complete list of model parameters, including clear and precise descriptions of the meaning of each parameter, together with the values or ranges for each, with justification or the primary source cited and important caveats about the use of these values noted.

* Please provide a clear statement about how the model was fitted to the data, including goodness-of-fit measure, the numerical algorithm used, which parameter varied, constraints imposed on parameter values, and starting conditions.

* For uncertainty analyses, please state the sources of uncertainties quantified and not quantified [can include parameter, data, and model structure].

* Please provide sensitivity analyses to identify which parameter values are most important in the model. Uncertainty estimates seek to derive a range of credible results on the basis of an exploration of the range of reasonable parameter values. The choice of method should be presented and justified.

* Please discuss the scientific rationale for the choice of model structure and identify points where this choice could influence conclusions drawn. Please also describe the strength of the scientific basis underlying the key model assumptions.

* For studies that develop a prediction model or evaluate its performance, please ensure that the study is reported according to the TRIPOD statement (https://www.equator-network.org/reporting-guidelines/tripod-statement) and include the completed checklist as Supporting Information. Please add the following statement, or similar, to the Methods: "This study is reported as per the Transparent Reporting of a Multivariable Prediction Model for Individual Prognosis Or Diagnosis (TRIPOD) statement (S1 Checklist)." For studies using machine learning, please use the TRIPOD-AI checklist. When completing the checklist, please use section and paragraph numbers, rather than page numbers.

---

## [Decision Letter · Decision Letter 2]

7 Jul 2025

Dear Dr. Bousema,

Thank you very much for re-submitting your manuscript "The transmission blocking activity of artemisinin-combination, non-artemisinin, and 8-aminoquinoline antimalarial therapies: a pooled analysis of individual participant data." (PMEDICINE-D-25-00573R2) for review by PLOS Medicine.

I have discussed the paper with my colleagues and the academic editor and it was also seen again by the statistical reviewer. I am pleased to say that provided the remaining editorial and production issues are dealt with we are planning to accept the paper for publication in the journal.

[LINK]

We look forward to receiving the revised manuscript by Jul 14 2025 11:59PM.   

Sincerely,

Alison

Alison Farrell, Ph.D.

Senior Editor 

PLOS Medicine

plosmedicine.org

Requests from Editors:

* In The Abstract:

Please use active voice (here and throughout)

please qualify the analysis as post hoc or retrospective on line 40.

please specify the trials are of antimalarials on line 43/44

please remove ‘(NMA)’ from the abstract, line 51 (abbreviation can be stated in main text but is unnecessary in the Abstract as is only mentioned once). Similarly is it necessary to abbreviate the antimalarials in the Abstract given that the abbreviations are not used again in the Abstract? Please comment.

please clarify what is meant by ‘but effective response compared to” (lines 62, 63). Was there a or no significant difference?

* In the author summary, in the final bullet point of 'What Do These Findings Mean?', please include the main limitations of the study in non-technical language.

* In Figure legends, when a p value is given, please specify the statistical test used to determine it.

* In Figures, please show graph axes beginning at zero. If this is not possible, please show a break in the axis.

* Please lighten the colour scheme in Figure 5 as the black font is difficult to read on the dark grey shading.

* In the Kaplan-Meier curve(s) please provide the number at risk for each time interval. Please ensure that the data included in the curves is presented in a supplementary table or Excel spreadsheet.

* In the Introduction, please very briefly explain what a mosquito feeding assay is and measures, line 124, for clarity for the general reader.

* Please clarify in the methods the consent for the clinical studies so that readers don’t need to go to the primary sources—was written informed consent obtained for all study participants?

* Line 303—please define DMFA

* Please ensure that all antimalarial abbreviations are spelled out in the main text (i.e. after the Abstract, even if used in the Abstract).

* Line 355—microscopic rather than microscopical?

* Line 412—abrogated rather than annulled?

* Please check for use of patient-centered language. Please note that patient-centered language is constructed with the use of post-modified nouns (e.g. 'patients with psoriasis’ (or similar) instead of ‘psoriasis patients’) putting the person first in the sentence structure. Please consider whether use of ‘carriers’ is patient-centered.

* Lines 450, 488—qualify claim of primacy with ‘to our knowledge’.

* Last paragraph of Discussion—please find a synonym/rephrase one of the uses of ‘confirm’. Paragraph should focus on the novel findings of this manuscript, unless the confirmation of prior findings sheds new light/points in a new direction, in which case please discuss that aspect of the work.

* Please include the appropriate reporting checklist for the NMA and include it as supplementary information.

Comments from Reviewers:

Reviewer #2: The authors have addressed all my points

Michael Dewey

[LINK]

---

## [Editor Report · Decision Letter 3]

14 Jul 2025

Dear Dr Bousema, 

On behalf of my colleagues and the Academic Editor, Lorenz von Seidlein, I am pleased to inform you that we have agreed to publish your manuscript "The transmission blocking activity of artemisinin-combination, non-artemisinin, and 8-aminoquinoline antimalarial therapies: a pooled analysis of individual participant data." (PMEDICINE-D-25-00573R3) in PLOS Medicine.

PRESS

Sincerely, 

Alison

Alison Farrell, Ph.D. 

Senior Editor 

PLOS Medicine